# An Enhanced Lightweight Dynamic Pseudonym Identity Based Authentication and Key Agreement Scheme Using Wireless Sensor Networks for Agriculture Monitoring

**DOI:** 10.3390/s19051146

**Published:** 2019-03-06

**Authors:** Meriske Chen, Tian-Fu Lee, Jiann-I Pan

**Affiliations:** 1Institute of Medical Sciences, Tzu Chi University, No. 701, Zhongyang Road, Sec. 3, Hualien 97004, Taiwan; 104325120@gms.tcu.edu.tw; 2Department of Medical Informatics, Tzu Chi University, No. 701, Zhongyang Road, Sec. 3, Hualien 97004, Taiwan; jipan@mail.tcu.edu.tw

**Keywords:** agriculture monitoring, agriculture WSN, key agreement, dynamic identity, agriculture decision support system, lightweight authentication

## Abstract

Agriculture plays an important role for many countries. It provides raw materials for food and provides large employment opportunities for people in the country, especially for countries with a dense population. To enhance agriculture productivity, modern technology such as wireless sensor networks (WSNs) can be utilized to help in monitoring important parameters in thw agricultural field such as temperature, light, soil moisture, etc. During the monitoring process, if security compromises happen, such as interception or modification of the parameters, it may lead to false decisions and bring damage to agriculture productivity. Therefore, it is very important to develop secure authentication and key agreement for the system. Recently, Ali et al. proposed an authentication and key agreement scheme using WSNs for agriculture monitoring. However, it fails to provide user untraceability, user anonymity, and session key security; it suffers from sensor node impersonation attack and perfect forward secrecy attack; and even worse has denial of service as a service. This study discusses these limitations and proposes a new secure and more efficient authentication and key agreement scheme for agriculture monitoring using WSNs. The proposed scheme utilizes dynamic pseudonym identity to guarantee user privacy and eliminates redundant computations to enhance efficiency.

## 1. Introduction

Agriculture plays an important role for many countries around the world. In some countries, agriculture is not only essential to provide food and raw material supply for its citizen, but also to provide large employment opportunities for its people. In some dense populated countries like India [1], Nigeria [2], and Pakistan [3], agriculture even becomes their economic backbone. Since agriculture is essential for life, when the world population is rising, the demand for agriculture products is also increasing and if there is no improvement in agriculture production, someday people may face challenges about food availability. 

Food availability depends on crops productivity and many other diverse factors such as livestock, labor, sophisticated machines, etc. While other diverse factors such as livestock, labor, climates, soils, tools, and technology vary from country to country or even from farm to farm [4]; the factors related to crop productivity almost remain similar anywhere, such as whether the farms have enough water, fertilizer, temperature, light, etc. On the other hand, farmers are also facing many challenges such as labor shortage, natural disaster (drought, flood, typhoon, etc.), land degradation, water availability, climate change, or any other stressors which might bring a decline in crop productivity. If there is no innovation to maintain or even increasing crop productivity, it is possible that someday the world might suffer from food supply shortage. If this happen, it might bring chaos in many countries and people may suffer from poverty, malnutrition, and hunger. Therefore, a good agriculture system to help in managing and facing those many challenges are important. For example, to make sure the crop succeeds, physical or environmental parameters such as temperature, luminance, humidity, wind direction, wind speed, moisture, acidity, water level, pollutant, etc. need to be routinely and constantly monitored. To make crop productivity sustainable, integrating traditional farming methods and information technology such as Wireless Sensor Networks (WSNs) in agriculture can help this necessity. Applications used for WSNs in agriculture may use different kinds of sensor devices to ones in healthcare, forests, urban areas, and the military since their monitoring range, functionality, and power supply status are different. Additionally, wireless sensor network technology for agriculture monitoring can monitor farm temperature, humidity, light, carbon dioxide, soil moisture, acidity, pests, etc., and can be used for plant epidemic monitoring and early warning systems. For the deployment, the sensors in the forest and military must consider the terrain such as rivers, valleys, etc. that are irregular. The distribution of urban sensors must consider factors such as roads and buildings. The sensors for the human body must consider body shape and portability. The sensors in farms are often arranged in rows due to the arrangement of crops. In terms of function, the sensors in forests and farms sense temperature, humidity, light, carbon dioxide, soil moisture, acidity, pests, etc. The urban sensors sense dust, air pollution, temperature, humidity, etc. In addition to sensing sound and images, sensors in the military must sometimes be able to sense toxic or chemical substances. 

WSN is a network infrastructure formed by a large number of sensor nodes to wirelessly monitor physical or environmental parameters. It can constantly monitor those important parameters in agriculture and instantly give clear notifications to user/agriculture professionals if some abnormal conditions are found. Therefore, WSNs can be used to manage or even improved crop productivity by monitoring important parameters in agriculture fields that affect the growth of the crops.

Figure 1 shows the structure of agriculture monitoring system using WSNs. Sensor nodes are manually deployed in the agriculture field and then routinely collect the data from physical or environmental parameters. First, these sensor nodes automatically collect data from physical or environmental parameter in the field and then send the collected data to the gateway node via wireless technology. The gateway node then sends the collected data to the user/agriculture professional via Internet. The user/agriculture professional will use these collected data as a basis to help them in decision making. Any misleading data such as false command, data modification, or wrong parameters may lead to a false decision which in turn could bring damage to crop productivity, such as crop failure and low return. It means that these collected data need to be protected from illegal access or data modification since they play an important role in agriculture decision support systems and are very important to help user/agriculture professionals in decision making. To accommodate this purpose, a clear mechanism about how the collected data are exchanged between legal participants is needed. This study proposes a secure and efficient authentication and key agreement scheme in agriculture monitoring system to meet these necessities.

### 1.1. Literature Reviews

Agriculture monitoring will help farmers to optimize their natural or artificial resources in their agricultural activities, which will influence their crop productivity. Some initial researches about the framework for agriculture monitoring based on WSNs [5,6,7,8] give important background about WSNs utilization in agriculture, especially about how this system can help decision support systems through better monitoring of their agriculture field. For example, Luis, et al. [5] gave a review about wireless sensor technologies for the agriculture and food industry; Jiber, et al. [6] and Anurag, et al. [7] presented a precision agriculture monitoring framework using WSNs; and Panchard, et al. [8] showed how wireless sensor technology can be used to help farming decision support. However, those existing frameworks do not explain about how those particular participants are authenticated between each other. 

In recent years, more and more researchers proposed an authentication and key-agreement scheme for WSNs environment [9,10,11,12,13,14,15,16,17,18,19,20,21,22,23,24,25,26]. Most of them proposed schemes for general purposes [9,10,11,12,13,14,15,17,24] and few of them proposed schemes for specific purposes [16,18,19,20,25,26]. For example, in 2009, Pecori and Veltri [25] proposed a new alternative key agreement protocol for setting up multimedia sessions between user agents (UAs) without requiring any pre-shared key or trust relationship or PKI, and it has been implemented and integrated in a publicly available VoIP UA. In 2012, Pecori [26] developed a new protocol for establishing a security association between two peers willing to set up a VoIP or multimedia communication through the standard SIP protocol. The proposed protocol is based on the MIKEY protocol and the Diffie-Hellman algorithm for key establishment, and allows the authentication via peer certificates without using any centralized PKI. In the same year, Das, et al. [9] proposed a dynamic password-based user authentication scheme for large-scale hierarchical WSNs. It consists of three entities which are the user, base station, and cluster head. Then, in 2013, Xue et al. [10] proposed a temporal-credential-based for WSNs and Shi et al. [11] proposed a new user authentication protocol using elliptic curves cryptography for WSNs. In 2015, there was even a study about group key management for WSNs [13]. Followed these studies, there were Li et al. [12] and He et al. [15] whose showed weaknesses of Xue et al.’s scheme [10] and both of them then proposed an improved scheme. In 2015, Lee [14] showed weaknesses of Li et al.’s scheme [12] and then proposed an improved scheme using extended chaotic maps. In the same year, Mesit and Brusta [24] proposed a secured node-to-node key agreement protocol, whose shared key is based on a symmetric encryption algorithm to solve the resource-constrained problem. Moreover, in 2016, Kumari et al. [17] mentioned weaknesses of both Li et al.’s scheme [12] and He et al.’s scheme [15] and then proposed an improved scheme using chaotic maps. 

Fewer researchers discuss about authentication and key agreement scheme using WSNs for specific purposes, for example, WSNs for healthcare through body sensor networks [16,18,19,20], WSNs for military [21] or multimedia [22] or agriculture monitoring [23]. 

### 1.2. Motivation and Contributions

The importance of modern technology utilization in agriculture is already described in above. It also followed by how essential a secure and efficient user authentication and key-agreement scheme for agriculture monitoring using WSNs. 

Dynamic pseudonym identity schemes [27,28,29], which were used by both Ali et al.’s scheme [23] and the proposed scheme, are quite popular and widely used in many security researches area. Dynamic pseudonym identity means that the transaction uses anonym identity and that the specific anonym identity dynamically changes in every new transaction. Anonymity is important in the agriculture area because it provides legitimate users with protection of their real identities. In the agriculture environment, we can assume that sensor nodes are put openly in the field. If a system does not provide anonymity, an attacker who targeting a particular participant can easily distinguish a transaction belongs to whom. Then he/she is able to perform attacks to his/her particular target. For example, Alice is an epidemic specialist and works on a farm. An adversary who tries to harm the farm facilities obtains Alice’s identity and knows that she is responsible for assisting in monitoring the farm’s temperature, humidity, and pests. The adversary may perform social engineering or dictionary attacks to obtain Alice’s password or login information, and then can log in to the system for agriculture monitoring to tamper with information and damage facilities. Therefore, by using dynamic pseudonym identity, the scheme is expected to be able to provide un-traceability, privacy and user anonymity to its user. However, Ali et al.’s scheme fails to provide user anonymity and user un-traceability. It also suffered from other severe security compromises such as insider attack, sensor node attack, perfect forward secrecy, and session key security. Moreover, Ali et al.’s scheme even suffered from denial of service which happened after a user/agriculture professional has successfully updated their password. 

The rest of this study is organized as follows. Section 2 reviews Ali et al.’s scheme and discusses the detail of security weaknesses in Ali et al.’s scheme. Section 3 presents the proposed scheme. Section 4 presents security analysis of the proposed scheme. Section 5 analyzes security and performances comparisons with Ali et al.’s scheme. Finally, Section 6 draws conclusions.

## 2. Preliminary

Although this study discusses the weaknesses of Ali et al.’s scheme, this study also recognizes the importance and advantages of their scheme, especially because of the novelty of their study. This study also followed their architecture for agriculture monitoring using WSNs, also utilizes dynamic pseudonym identity and three-factor-security, which are similar with Ali et al.’s scheme. This section consists of three sub-sections which discuss about the importance and advantage of Ali et al.’s scheme, Ali et al.’s scheme, and the weaknesses of Ali et al.’s scheme.

The notations used in Ali et al.’s scheme and in the proposed scheme are elaborated in Table 1.

### 2.1. The Importance and Advantage of Ali et al.’s Scheme

Ali et al. proposed a novel authentication and key agreement scheme using WSNs for agriculture monitoring. At first, they mentioned about how important agriculture is for economic systems and how WSNs technology can be utilized to face many challenges that exist in agriculture. Then, they reviewed some literature that related to security in the WSNs environment and summarized security requirements that need to be fulfilled in a scheme. Then, they presented their scheme, the security analysis and the performance evaluation of their scheme. 

Compare with other existing WSNs schemes where most of them consist of three entities, Ali et al.’s scheme consist of four entities instead, which are the user/agriculture professional, base station BS, sensor node, and gateway node. The BS acts as system administrator and becomes the central entity to authenticate other entities. Without BS, other entities will never have the chance to truly trust each other in the authentication and key agreement scheme.

### 2.2. Ali et al.’s Scheme

In 2017, Ali et al. [23] proposed a WSNs scheme for agriculture monitoring, which consists of system setup phase; user/agriculture professional registration phase; login, authentication, and session key agreement phase; password update phase; and dynamic node addition phase.

#### 2.2.1. System Setup Phase

To initialize the organization, the system administrator SA selects distinct identity $ID_{SN_{j}}$ for m sensor node $SN_{j}$, where 1≤j≤m and also selects distinct identity $ID_{GWN_{j}}$ for each gateway node $GWN_{j}$. SA computes the shared key ${\mathrm{RI}}_{j}=h\left({\mathrm{ID}}_{{\mathrm{SN}}_{j}}\|X\right)$ for $SN_{j}$, where X is the secret key of the base station BS and computes the shared key $h(X_{\mathrm{BS}-{\mathrm{GWN}}_{j}})$ for ${\mathrm{GWN}}_{j}$. Finally, SA keeps $\left\{{\mathrm{RI}}_{j},{\;\mathrm{ID}}_{{\mathrm{SN}}_{j}}\right\}$ into $SN_{j}$’s memory and keeps {$h(X_{\mathrm{BS}-{\mathrm{GWN}}_{j}})$, ${\mathrm{ID}}_{{\mathrm{GWN}}_{j}}$} into ${\mathrm{GWN}}_{j}$’s memory. Then, SA deploys each sensor node $SN_{j}$ and ${\mathrm{GWN}}_{j}$ in a target area. Here, the SA acts as BS representative to initialize the identity and the shared key with ${\mathrm{SN}}_{j}$ and ${\mathrm{GWN}}_{j}$. 

#### 2.2.2. User/Agriculture Professional Registration Phase

As shown in Figure 2, user/agriculture professional $U_{i}$ needs to register to the base station BS. The following steps were executed when $U_{i}$ want to become a legitimate user in this agriculture monitoring system.
Step 1:The $U_{i}$ selects his/her own identity $ID_{i}$, password $PW_{i}$ and imprints biometric $F_{i}$ on the sensor device and then computes $Gen\left(F_{i}\right)=\left(X_{F},\;P_{F}\right)$, $RPW_{i}=h\left(PW_{i}\|X_{F}\right)$, where Gen(.) is a generate function of fuzzy extractor and $\left(X_{F},\;P_{F}\right)$ are, respectively, secret and public keys. Now, $U_{i}$ sends $\left\{ID_{i},\;RPW_{i}\right\}$ to BS via trustworthy channel.Step 2:When obtained the registration request from $U_{i}$, BS firstly calculates $A_{i}=h\left(ID_{i}\|X\right)$, $B_{i}=A_{i}⊕h\left(RPW_{i}\|ID_{i}\right)$, $C_{i}=A_{i}⊕h\left(B_{i}\|X\right)$ and $D_{i}=h\left(A_{i}\|\;RPW_{i}\|ID_{i}\right).$ Afterwards, BS issues a smartcard having parameters, i.e., $\left\{B_{i},\;C_{i},\;D_{i},\;h\left(.\right)\right\}$ and sends it to $U_{i}$ via the same channel.Step 3:After obtaining the smartcard from BS, $U_{i}$ embeds $P_{F}$ and Gen(.) in the memory of smartcard, i.e., $\left\{B_{i},\;C_{i},\;D_{i},\;h\left(.\right),\;P_{F},\;Gen(.)\right\}$.

#### 2.2.3. Login Phase

When a user/agriculture professional $U_{i}$ wants to know the environmental information such as temperature, light, humidity, soil etc., he/she has to login to access these information. As shown in Figure 3, the following steps were executed to accomplish this login phase.
Step 1:The $U_{i}$ inserts his/her own smartcard into card reader and inputs $ID_{i}$, $PW_{i}$ and also imprints $F_{i}$ on a sensor device. Now, the card reader computes $Rep(F_{i}$, $P_{F})$ = $X_{F}^{*}$, $RPW_{i}^{*}$ = $h\left(PW_{i}\|X_{F}^{*}\right)$, $A_{i}^{*}$ = $B_{i}$ ⊕ $h(RPW_{i}^{*}\;\|\;ID_{i})$, $D_{i}^{*}$ = $h(A_{i}^{*}\;\|\;RPW_{i}^{*}\;\|\;ID_{i})$, ${\left[h(B_{i}\|X)\right]}^{*}$ = $C_{i}\;$⊕$\;A_{i}^{*}$ and verifies if $D_{i}^{*}$ equals $D_{i}$. If this verification holds then the system continues the process. Otherwise, the session is terminated.Step 2:Now, $U_{i}$ generates a random nonce $R_{U}$ and enumerates $DID_{i}=ID_{i}⊕h(B_{i}\|\;X)$, $M_{1}=E_{Ai}\left(R_{U}\|ID_{SN_{j}}\|ID_{GWN_{j}}\|T_{1}\right)$, $M_{2}=h\left(R_{U}\|ID_{i}\|T_{1}\|\;h\left(B_{i}\|X\right)\right)$ and sends $\left\{B_{i},\;DID_{i},\;M_{1},\;M_{2}\right\}$ to BS via public channel.

#### 2.2.4. Authentication and Session Key Agreement Phase

As shown in Figure 3, after login phase is successfully authenticated, the authentication and session-key phase were executed in the following steps.
Step 1:Upon obtaining the message $\left\{B_{i},\;DID_{i},\;M_{1},\;M_{2}\right\}$ from $U_{i}$, the BS computes $ID_{i}^{*}=DID_{i}⊕h(B_{i}\|\;X)$, $\left(R_{U}^{*}\|ID_{SN_{j}}^{*}\|\;ID_{GWN_{j}}^{*}\|T_{1}\right)=D_{Ai}\left(M_{1}\right)$ and checks if $T_{2}-T_{1}\leq ∆T$ holds. If this does not true, then session expires. Otherwise, BS computes $M_{2}^{*}=h\left(R_{U}^{*}\|ID_{i}^{*}\|T_{1}^{*}\|h(B_{i}\|X\right))$ and verifies if $M_{2}^{*}$ equals $M_{2}$ or not. If it holds, then $U_{i}$ is legal and BS goes to next step. Otherwise, the session is rejected.Step 2:Now, the BS produces a random nonce $R_{BS}$ and computes $M_{3}=E_{h(X_{BS-GWN_{j}})}\left(ID_{i}\|RI_{j}\|ID_{SN_{j}}\|R_{U}\|R_{BS}\|T_{3}\right)$, $M_{5}=h\left(M_{2}\|ID_{i}\|T_{3}\|R_{BS}\|R_{U}\right)$ and then sends $\left\{M_{3},\;M_{2},\;M_{5}\right\}$ to $GWN_{j}$ via public channel.Step 3:After getting request message $\left\{M_{3},\;M_{2},\;M_{5}\right\}$ from BS, the $GWN_{j}$ computes $\left(ID_{i}\|RI_{j}\|ID_{SN_{j}}\|R_{U}\|R_{BS}\|T_{3}\right)=D_{h(X_{BS-GWN_{j}})}\left(M_{3}\right)$ and $\left(R_{U}^{*}\|ID_{SN_{j}}^{*}\|\;ID_{GWN_{j}}^{*}\|\;T_{1}\right)=\;D_{A_{i}}\left(M_{1}\right)$, then checks if two condition $T_{4}-T_{3}\leq ∆T$ and $M_{5}^{*}{\;}_{=}^{?}\;M_{5}$ hold. If both conditions are true then it proceeds further. Otherwise, the session is terminated.Step 4:Now, the $GWN_{j}$ generates a random nonce $R_{GWN_{j}}$ and calculates $M_{6}=\;E_{RI_{j}}\left(R_{BS}\|T_{5}\|R_{U}\|R_{GWN_{j}}\|ID_{i}\right)$, $M_{7}=h\left(M_{2}\|RI_{j}\|R_{GWN_{j}}\|ID_{i}\|R_{U}\right)$ and then sends $\left\{\;M_{2},\;\;M_{6},\;\;M_{7}\;\right\}$ to $SN_{j}$.Step 5:Upon obtaining the message from $GWN_{j}$, the $SN_{j}$ computes $\left(R_{BS}\|T_{5}\|R_{U}\|R_{GWN_{j}}\|\;ID_{i}\right)=D_{RI_{j}}\left(M_{6}\right)$, $M_{7}^{*}=h\left(M_{2}\|RI_{j}\|R_{GWN_{j}}\|\;ID_{i}\|R_{U}\right)$ and then verifies if two conditions $T_{6}-T_{5}\leq ∆T$ and $M_{7}^{*}{\;}_{=}^{?}\;M_{7}$ hold. If both are true, then $SN_{j}$ goes to the next step, Otherwise, the session is terminated.Step 6:Now, the $SN_{j}$ generates a random nonce $R_{SN_{j}}$, computes $M_{8}=R_{SN_{j}}⊕h\left(RI_{j}\|R_{GWN_{j}}\right)$, $SK=h\left(R_{GWN_{j}}\|\;R_{U}\|R_{SN_{j}}\|\;RI_{j}\|M_{2}\right)$, $M_{9}=h\left(SK\|ID_{i}\right)$ and sends $\left\{\;M_{8},\;\;M_{9},\;\;T_{7}\;\right\}$ to $GWN_{j}$ via public channel.Step 7:After getting the message from $SN_{j}$, $GWN_{j}$ firstly verifies if $T_{8}-T_{7}\leq ∆T$ holds. If true, the process continues. Otherwise, the session expires. Then, $GWN_{j}$ calculates $R_{SN_{j}}^{*}=M_{8}⊕h\left(RI_{j}\|R_{GWN_{j}}\right)$, $SK^{*}=h\left(R_{GWN_{j}}\|\;R_{U}\|R_{SN_{j}}^{*}\|\;RI_{j}\|M_{2}\right)$ and $M_{9}^{*}=h\left(SK^{*}\|ID_{i}\right)$, then checks if $M_{9}^{*}{\;}_{=}^{?}\;M_{9}$. If it holds, the next step proceeds. Otherwise, the session is terminated.Step 8:The $GWN_{j}$ computes $M_{10}=\;E_{h\left(R_{U}\|ID_{i}\right)}\left(RI_{j}\|T_{9}\|R_{GWN_{j}}\|\;R_{SN_{j}}\|\;M_{2}\right)$ and sends $\left\{\;M_{9},\;\;M_{10}\;\right\}$ to $U_{i}$ via public channel.Step 9:After getting the message from $GWN_{j}$, $U_{i}$ computes $\left(RI_{j}\|T_{9\;}\|\;R_{GWN_{j}}\|\;R_{SN_{j}}\|\;M_{2}\right)=D_{h\left(R_{U}\|ID_{i}\right)}(M_{10})$ and verifies if $T_{10}-T_{9}\leq ∆T$ holds. If it holds, the next step proceeds.Step 10:The $U_{i}$ calculates $SK^{*}$=$h\left(R_{GWN_{j}}\|R_{U}\|R_{SN_{j}}\|RI_{j}\|M_{2}\right)$, $M_{9}^{*}=h\left(SK^{*}\|\;ID_{i}\right)$ and checks if $M_{9}^{*}{\;}_{=}^{?}\;M_{9}$ holds. If it holds, mutual-authentication and session-key agreement holds.

#### 2.2.5. Password Updates or Change Phase

In Ali et al.’s password update or change phase, user $U_{i}$ modifies his/her password without intervention with the base station. As shown in Figure 4, the following steps were executed to update or change password.
Step 1:The $U_{i}$ inserts his/her own smartcard into the card reader and enters $ID_{i}$, $PW_{i}$ and imprints $F_{i}$ on a sensor device. Now, the card reader computes $Rep(F_{i}$, $P_{F})$ = $X_{F}^{*}$, $RPW_{i}^{*}=$
$h\left(PW_{i}\|X_{F}^{*}\right)$, $A_{i}^{*}=\;B_{i}$⊕ $h(RPW_{i}^{*}\;\|\;ID_{i})$, $D_{i}^{*}=$
$h(A_{i}^{*}\;\|\;RPW_{i}^{*}\;\|\;ID_{i})$, ${\left[h(B_{i}\|X)\right]}^{*}$ = $C_{i}\;$⊕$\;A_{i}^{*}$ and verifies if $D_{i}^{*}{\;}_{=}^{?}\;D_{i}$. If this verification holds, then continues the process. Otherwise, the session is terminated.Step 2:The $U_{i}$ enters new password $PW_{i}^{new}$ and computes $RPW_{i}^{new}=h\left(PW_{i}^{new}\|X_{F}\right)$, $B_{i}^{new}=B_{i}$⊕$h\left(RPW_{i}\|ID_{i}\right)$⊕$h\left(RPW_{i}^{new}\|ID_{i}\right)$, $A_{i}^{new}=B_{i}^{new}⊕\left(RPW_{i}^{new}\|ID_{i}\right)$, $C_{i}^{new}=C_{i}⊕A_{i}⊕A_{i}^{new}$ and $D_{i}^{new}=h(A_{i}^{new}\|RPW_{i}^{new}\|ID_{i})$. Then, $\left\{B_{i},\;C_{i},\;D_{i}\right\}$ are replaced with $\left\{B_{i}^{new},\;C_{i}^{new},D_{i}^{new}\right\}\;\mathrm{respectively}$.

#### 2.2.6. Dynamic Node Addition Phase

This phase was used to add, replace, or drop a sensor node in the field. Let $S_{n}\;$ becomes a sensor node that will be added into the field. SA chooses $ID_{n}$ of $S_{n}$, calculates $RI_{n}=h\left(ID_{n}\|X\right)$ and keeps $\left\{RI_{n},\;ID_{n}\right\}$ into sensor nodes memory. At last, SA deploys $S_{n}$ to the field.

### 2.3. Weaknesses of Ali et al.’s Scheme

This section discusses the weaknesses of Ali et al.’s scheme in detail. Ali et al.’s scheme weaknesses are divided into three sections which are violation of traceability, insider attack, and denial of service as a service. For the insider attack, it is divided into four other sub-sections which are violation of user anonymity, sensor node impersonation attack, perfect forward secrecy, and violation of session key security. The details are described as follows.

#### 2.3.1. Violation of User Traceability

User traceability means the ability to distinguish if any transactions belong to or came from a certain user. Ali et al.’s scheme was trying to protect users’ real identity by using pseudonym identity $DID_{i}$, where $DID_{i}$= $ID_{i}⊕h(B_{i}\|\;X)$. However, the value of $DID_{i}$ is constant in every transaction. By using or checking the $DID_{i}$, the adversary is able to distinguish existing transactions easily whether they are generated from the same user or not. Since the transaction is easily be distinguished, therefore, the scheme of Ali et al. fails to provide user un-traceability.

#### 2.3.2. Insider Attack

Insider attack happens when a malicious legal participant successfully captures key values of others, such as a shared key, and then uses that key to launch some security violations or attacks. In Ali et al.’s scheme, each sensor node $SN_{j}$ has a shared key $RI_{j}$ with base station BS, where $RI_{j}=h\left(ID_{SN_{j}}\|\;X\right)$ and it should be known only by BS and $SN_{j}$. But, other legal participants such as $GWN_{j}$ and $U_{i}$ can also obtain $RI_{j}$ automatically from a legal transaction during the authentication and session key agreement phase. $GWN_{j}$ and $U_{i}$ obtain $RI_{j}$ when they decrypt $D_{A}\left(M_{1}\right)$ and $D_{h\left(R_{U}\|ID_{i}\right)}\left(M_{10}\right)$, respectively, where $M_{1}\;$= $E_{Ai}\left(R_{U}\;\|\;ID_{SN_{j}}\|\;ID_{GWN_{j}}\|\;T_{1}\right)$ and $M_{10}=E_{h\left(RU\|IDi\right)}\left(RI_{j}\;\|\;T_{9}\;\|\;R_{GWN_{j}}\|\;R_{SN_{j}}\|\;M_{2}\right)$. After these legal $GWN_{j}$ and $U_{i}$ obtain $RI_{j}$, they can use $RI_{j}$ to release some security violations or attacks such as sensor node capture attacks, impersonation attacks, and perfect forward secrecy attacks.

##### Violation of User Anonymity

User anonymity is important since it protects the real identity $ID_{i}$ of a user $U_{i}$ and ensures his/her privacy. In Ali et al.’s scheme, once a legal participant obtained a shared key $RI_{j}$, he/she can catch others’ existing transactions $\left\{M_{2},\;M_{6},\;M_{7}\right\}$ from the public channel, use $RI_{j}$ to decrypt $M_{6}$, and then get the $ID_{i}$ of $U_{i}$, where $D_{RI_{j}}\left(M_{6}\right)=\left(R_{BS}\;\|\;T_{5}\;\|\;R_{U}\;\|\;R_{GWN_{j}}\|\;ID_{i}\right)$. Therefore, the proposed scheme fails to provide user anonymity.

##### Sensor Node Impersonation Attack

Sensor node impersonation attack occurred when a malicious insider successfully acts as a legitimate sensor node. When a legitimate sensor node is breached or captured by an adversary, it might result in severe security breaches [30], such as eavesdropping, node malfunctioning, denial of service, node subversion, node outage, message corruption, false nodes, and node replication. In Ali et al.’s scheme, when a malicious user $U_{adv}$ or gateway node $GWN_{adv}$ tries to impersonate a sensor node $SN_{j}$ by using the shared key $RI_{j}$, first they catch the request message $\left\{M_{2},\;M_{6},\;M_{7}\right\}$ and then decrypt $M_{6}$, such as shown in previous subsection Violation of user anonymity. After that, they generate a timestamp $T_{7}$, a random nonce $R_{SN_{j}}$ and then compute $M_{8}$, SK and $M_{9}$, where $M_{8}=R_{SN_{j}}⊕h\left(RI_{j}\|\;R_{GWN_{j}}\right)$, $SK=h\left(R_{GWN_{j}}\;\|\;R_{U}\;\|\;R_{SN_{j}}\;\|\;RI_{j}\;\|\;M_{2}\right)$, $M_{9}=h\left(SK\;\|\;ID_{i}\right)$. Then, he/she sends {$\;M_{8},\;\;M_{9},\;\;T_{7}$} to $GWN_{j}$ and $GWN_{j}$ will send it to the user. Since both the key and procedure are true during computation, both $U_{i}$ and $GWN_{j}$ will not find any suspicious activity and will trust that malicious $SN_{j}$. Therefore, Ali et al.’s scheme cannot withstand sensor node impersonation attack.

##### Perfect Forward Secrecy Attack

A perfect forward secrecy attack occurs when an adversary can successfully obtain previous session keys by using a compromised key. In Ali et al.’s scheme, a malicious user $U_{adv}$ or gateway node $GWN_{adv}$ tries to generate previous session key SK by using known shared key $RI_{j}$. First, he/she obtains $R_{U}$ and $ID_{i}$ through $M_{6}$, such as shown in in previous subsection Violation of user anonymity. Then, using $R_{U}$ and $ID_{i}$, he/she decrypts $M_{10}$, where $\left(RI_{j}\|T_{9}\|R_{GWN_{j}}\|R_{SN_{j}}\|M_{2}\right)=D_{h\left(R_{U}\|ID_{i}\right)}(M_{10})$. Then, he/she calculates $SK^{*}$ and $M_{9}^{*}$, respectively, where $SK^{*}$ = $h\left(R_{GWN_{j}}\|\;R_{U}\;\|\;R_{SN_{j}}\|\;RI_{j}\;\|\;M_{2}\right)$ and $M_{9}^{*}=h\left(SK^{*}\;\|\;ID_{i}\right)$. To verify if $SK^{*}$ is true, the attacker compares $M_{9}^{*}$ with previous publicly known $M_{9}$ in {$\;M_{8},\;\;M_{9}\;,\;\;T_{7}$}. If equals, the adversary has confirmation that $SK^{*}$ is true. Therefore, Ali et al.’s scheme cannot withstand perfect forward secrecy attack.

##### Violation of Session Key Security

A session key is important to ensure the communication between legal participants in each session is secure. Violation of session key security happens when a non-legal participant can successfully generate a session key with other legal participants. In Ali et al.’s scheme, such as described in previous subsection Sensor node impersonation attack, a malicious insider successfully acts as a legitimate sensor node and is authenticated by a legal user $U_{i}$. When authentication and key agreement succeed, they will generate a session key and use that session key to communicate with each other. Therefore, Ali et al.’s scheme fails to provide session key security.

#### 2.3.3. Denial of Services as a Service in Authentication and Key Agreement Phase

Denial of Service as a Service (DoSaaS) happened when a service cannot continue to the next step simply because of the incompatibility procedures of the exchange scheme or because of false data calculation procedures in the scheme. In Ali et al.’s scheme, the denial of services as a service happens after user $U_{i}$ successfully updates his/her password.

In the update password phase, when $U_{i}$ wants to update his/her password, he/she first inserts his/her smart card and password, then inserts his/her new password $PW_{i}^{new}$. Then, $RPW_{i}^{new}$, $B_{i}^{new}$, $A_{i}^{new}$, $C_{i}^{new}$ and $D_{i}^{new}$ are computed, where $RPW_{i}^{new}=h\left(PW_{i}^{new}\|X_{F}\right)$, $B_{i}^{new}=B_{i}⊕h\left(RPW_{i}\|ID_{i}\right)⊕h\left(RPW_{i}^{new}\|ID_{i}\right)$, $A_{i}^{new}=B_{i}^{new}⊕h\left(RPW_{i}^{new}\;\|\;ID_{i}\right)$, $C_{i}^{new}=C_{i}⊕A_{i}⊕A_{i}^{new}$ and $D_{i}^{new}=h(A_{i}^{new}\|RPW_{i}^{new}\|ID_{i})$. At last, previous $\left\{B_{i},\;C_{i},\;D_{i}\right\}$ that were saved in the smart card are replaced with $\left\{B_{i}^{new},\;C_{i}^{new},D_{i}^{new}\right\}$, respectively. When $U_{i}$ wants to login after successfully updating his/her password, the login process fails due to denial of service. Details are explained below.

As shown in the login phase in Section 2.3, $U_{i}$ computes $\left[h(B_{i}\;\|\;X\right){]}^{*}$ = $C_{i}⊕A_{i}^{*}$, where $C_{i}$ is the new $C_{i}^{new}$ and $A_{i}^{*}$ is the new $A_{i}^{new}$, which means $\left[h(B_{i}\;\|\;X\right){]}^{*}$ = $C_{i}^{new}⊕A_{i}^{new}$. Unfortunately, in the update password phase, $C_{i}^{new}=C_{i}^{old}⊕A_{i}^{old}⊕A_{i}^{new}$, which means $C_{i}^{new}⊕A_{i}^{new}=\left(C_{i}^{old}⊕A_{i}^{old}⊕A_{i}^{new}\right)⊕A_{i}^{new}$= $C_{i}^{old}⊕A_{i}^{old}$= $\left[h(B_{i}\;\|\;X\right){]}^{old}$. Using this $\left[h(B_{i}\;\|\;X\right){]}^{old}$, $U_{i}$ computes $DID_{i}$, $M_{1}$ and $M_{2}$, where $DID_{i}$= $ID_{i}⊕h{\left(B_{i}\;\|\;X\right)}^{old}$, $M_{1}=\;E_{Ai}\left(R_{U}\;\|\;ID_{SN_{j}}\|\;ID_{GWN_{j}}\;\|\;T_{1}\right)$, $M_{2}=\;h\left(R_{U}\;\|\;ID_{i}\;\|\;T_{1}\|\;h{\left(B_{i}\|X\right)}^{old}\right)$, respectively, and send {$B_{i}^{new},DID_{i},M_{1},M_{2}\;$} to BS.

When BS get the request message {$B_{i}^{new},DID_{i},M_{1},M_{2}\;$} from $U_{i}$, BS will calculate $ID_{i}$ and $M_{2}$, where $ID_{i}^{*}$= $DID_{i}⊕h\left(B_{i}^{new}\|X\right)$ and $M_{2}^{*}=h(R_{U}\;\|ID_{i}^{*}\;\|\;T_{1}\|\;h\left(B_{i}^{new}X)\right)$. Then compare whether $M_{2}$ equals $M_{2}^{*}$. Since $M_{2}$ from $U_{i}$ was calculated by using $\left[h(B_{i}\;\|\;X\right){]}^{old}$ and $M_{2}^{*}$ from BS was calculated by using $h(B_{i}^{new}X)$, $M_{2}$ and $M_{2}^{*}$ will never be equal. When they do not equal, BS will reject the request message from $U_{i}$. Therefore, Ali et al.’s scheme suffers from DoSaaS.

## 3. Proposed Authentication and Key-Agreement Scheme Using WSNs for Agriculture Monitoring

The proposed scheme proposed some significant improvements compared to Ali et al.’s scheme. For example, to overcome violation of traceability in Ali et al.’s scheme, instead of using static $A_{i}$ and $DID_{i}$, the proposed scheme uses dynamic $A_{i}$ and $DID_{i}$. The proposed scheme also eliminates sensor node impersonation attack, perfect forward secrecy and violation of user anonymity by keeping the shared secret key $RI_{j}$ to be known only by BS and $SN_{j}$, while in Ali et al.’s scheme, the $RI_{j}$ is known by all participants. To overcome Denial of Service as a Service in Ali et al.’s scheme, the proposed scheme proposes a different structure for password update phase, where in order to complete the password update process, the user $U_{i}$ needs to send the new updated parameters to the base station *BS* to be processed. Moreover, to significantly improved efficiency, the proposed scheme only uses hash function in its computation, while Ali et al. used symmetric encryption-decryption for their scheme.

The proposed scheme consists of six phases, which are system setup phase; user/agriculture professional registration phase; login phase, authentication and session key agreement phase; password update or change phase; and dynamic node addition phase. Since the system setup phase and the dynamic node addition phase of the proposed scheme are similar with Ali et al.’s scheme, they are not presented here. Therefore, only user/agriculture professional registration phase; login phase; authentication and session key agreement phase; and password update or change phase are described in detail as follows. 

### 3.1. User/Agriculture Professional Registration Phase

In this phase, the user/agriculture professional $U_{i}$ registers to the base station BS. Each user $U_{i}$ has a SC which contains a pre-configured identity $ID_{i}{}^{pre}$ and a random number $r_{0}$. The pre-configured data is also stored in *BS*’s storage. The SC is transferred by using physical delivery. As shown in Figure 5, the following steps are executed to complete the registration phase.
Step 1:The $U_{i}$ selects his/her own identity $ID_{i}$, password $PW_{i}$, and imprints biometric $F_{i}$ on the sensor device and then computes $Gen\left(F_{i}\right)=\left(X_{F},\;P_{F}\right)$, $RPW_{i}=h\left(PW_{i}\|X_{F}\right)$, where Gen(.) is a generate function of fuzzy extractor and $\left(X_{F},\;P_{F}\right)$ are secret and public key respectively. Now, $U_{i}$ computes $REG_{i}=r_{0}⊕(ID_{i}\left||RPW_{i}\right||A_{i})$ and sends $\left\{\;ID_{i}{}^{pre},REG_{i}\right\}$ to BS.Step 2:When the registration request is received from $U_{i}$, if BS successfully verifies that ($ID_{i}{}^{pre},r_{0}$) is in BS’s storage and has not been registered, then BS computes $(ID_{i}\left||RPW_{i}\right||A_{i})=REG_{i}⊕r_{0}$, $B_{i}=h\left(A_{i}\|X\right)⊕h\left(ID_{i}\|RPW_{i}\right)$ and $D_{i}=\;h\left(A_{i}\;\|\;RPW_{i}\;\|\;ID_{i}\right)$. Afterwards, BS computes $RSP_{i}=h((ID_{i}||r_{0})⊕(B_{i}||D_{i})$ and sends {$RSP_{i}$} to $U_{i}$.Step 3:After receiving the response from BS, $U_{i}$ computes $\left(B_{i}||D_{i})=RSP_{i}⊕h((ID_{i}||r_{0}\right)$, and embeds $A_{i},\;B_{i},\;\;D_{i},\;h\left(.\right),\;P_{F}$ and Gen(.) in the memory of SC. 

### 3.2. Login Phase

When a user/agriculture professional $U_{i}$ wants to know the environmental information such as temperature, light, humidity, soil etc., he/she has to login to access these information. As shown in Figure 6, the following steps are executed to accomplish the login phase.
Step 1:The $U_{i}$ inserts his/her own smartcard into card reader, inputs $ID_{i}$, $PW_{i}$ and imprints his/her biometric $F_{i}$ on sensor device. Now, the card reader computes $Rep(F_{i}$, $P_{F})$ = $X_{F}^{*}$, $RPW_{i}^{*}$ = $h\left(PW_{i}\;\|\;X_{F}^{*}\right)$, ${\left[h(A_{i}\|X)\right]}^{*}$ = $B_{i}⊕h\left(ID_{i}\|RPW_{i}^{*}\right)$, $D_{i}^{*}\;$= $h(A_{i}^{*}\;\|\;RPW_{i}^{*}\;\|\;ID_{i})$ and verifies if $D_{i}^{*}$ equals $D_{i}$. If the verification holds, the system continues to process the request. Otherwise, the session is terminated.Step 2:Now, $U_{i}$ generates a random nonce $R_{U}$, computes $DID_{i}=\left(ID_{i}\|R_{U}\right)⊕h(h(A_{i}\|X)\|T_{1})$ and $M_{1}=h\left(R_{U}\|ID_{i}\|T_{1}\|h(A_{i}\|X\right))$, then send $\left\{A_{i},\;DID_{i},\;T_{1},\;M_{1},\;ID_{SN_{j}},\;ID_{GWN_{j}}\right\}$ to BS via public channel.

### 3.3. Authentication and Session Key Agreement Phase

As shown in Figure 7, after the $U_{i}$ is successfully authenticated in the login phase, the authentication and session-key agreement phase is executed as the following steps.
Step 1:Upon obtaining the message $\left\{A_{i},\;DID_{i},\;T_{1},\;M_{1},\;ID_{SN_{j}},\;ID_{GWN_{j}}\right\}$ from $U_{i}$, the BS checks if $T_{2}-T_{1}\leq ∆T$ holds. If this does not true then session expires. Otherwise, BS calculates $\left(ID_{i}\;\|\;R_{U}\right)=DID_{i}⊕h(h(A_{i}\|X)\;\|\;T_{1})$ and computes $M_{1}^{*}=h\left(R_{U}\;\|\;ID_{i}\;\|\;T_{1}\;\|\;h(A_{i}\|X\right))$, then verifies if $M_{1}^{*}$ equals $M_{1}$ or not. If this holds, BS goes to next step. Otherwise, the session is rejected.Step 2:Now, BS generates a random nonce $R_{BS}$ and computes a new $A_{i}^{new}$, where $A_{i}^{new}$=$h\left(R_{U}\|h\left(A_{i}\|X\right)\right).$ Then, BS computes $M_{2}$=$\;h\left(A_{i}^{new}\|X\right)$⊕$h\left(h\left(A_{i}\|X\right)\|R_{BS}\right)$, $M_{3}=\left(R_{U}\;\|\;R_{BS}\;\|\;ID_{i}\right)⊕h\left(X_{BS-GWN_{j}}\|\;T_{3}\right)$, $M_{4}=h\left(M_{1}\|M_{2}\|ID_{i}\|T_{3}\|R_{BS}\|R_{U}\right)$ and $M_{5}=R_{BS}$ ⊕ $h\left(h\left(ID_{SN_{j}}\|X\right)\;\|\;T_{3}\right)$ and sends $\left\{\;M_{1},\;\;M_{2},\;\;M_{3},\;M_{4},\;\;M_{5},\;T_{3}\right\}$ to $GWN_{j}$ via public channel.Step 3:After getting the request message from BS, the $GWN_{j}$ checks if $T_{4}-T_{3}\leq ∆T$ holds. If this does not true then session expires. Otherwise, $GWN_{j}$ calculates $\left(R_{U}\|R_{BS}\|ID_{i}\right)=M_{3}⊕h\left(X_{BS-GWN_{j}}\|\;T_{3}\right)$ and $M_{4}^{*}=h\left(M_{1}\|M_{2}\|ID_{i}\|T_{3}\|R_{BS}\|R_{U}\right)$, then checks if $M_{4}^{*}{\;}_{=}^{?}\;M_{4}$ holds. If the condition is true then it proceeds further. Otherwise, the session is terminated.Step 4:Now, the $GWN_{j}$ generates a random nonce $R_{GWN_{j}}$, calculates $M_{6}$= $\left(R_{U}\|R_{GWN_{j}}\|ID_{i}\right)⊕h\left(R_{BS}\|M_{5}\|T_{5}\right)$ and $M_{7}=h\left(M_{1}\|M_{2}\|R_{BS}\|R_{GWN_{j}}\|ID_{i}\|R_{U}\right)$, then sends $\left\{\;M_{1},\;\;M_{2},\;\;M_{5},\;\;M_{6},\;M_{7},\;T_{3},\;\;T_{5}\right\}$ to $SN_{j}$.Step 5:Upon obtaining the message from $GWN_{j}$, the $SN_{j}$ checks if $T_{6}-T_{5}\leq ∆T$ holds. If this does not true then session expires. Otherwise, $SN_{j}$ calculates $R_{BS}^{*}=M_{5}⊕h\left(RI_{j}\;\|\;T_{3}\right)$, $\left(R_{U}\;\|\;R_{GWN_{j}}\|\;ID_{i}\right)=M_{6}⊕h\left(R_{BS}^{*}\;\|\;M_{5}\;\|\;T_{5}\right)$ and $M_{7}^{*}=h\left(M_{1}\;\|\;M_{2}\;\|\;R_{BS}^{*}\;\|\;R_{GWN_{j}}\|\;ID_{i}\;\|\;R_{U}\right)$. Then, $SN_{j}$ verifies if $M_{7}^{*}{\;}_{=}^{?}\;M_{7}$ holds. If the condition is true then it proceeds further. Otherwise, the session is terminated.Step 6:Now, the $SN_{j}$ generates a random nonce $R_{SN_{j}}$, computes $M_{8}=R_{SN_{j}}⊕h\left(R_{GWN_{j}}\|\;R_{BS}^{*}\;\|\;T_{7}\right)$, $SK=h\left(R_{GWN_{j}}\|\;R_{U}\;\|\;R_{SN_{j}}\|\;R_{BS}^{*}\;\|\;ID_{i}\;\|\;M_{1}\right)$ and $M_{9}=h\left(SK\;\|\;R_{BS}^{*}\;\|\;R_{U\;}\|\;M_{2}\;\|\;T_{7}\right)$. Then, $SN_{j}$ sends $\left\{\;M_{1},\;\;M_{2},\;M_{8},\;\;M_{9},\;\;T_{7}\;\right\}$ to $GWN_{j}$.Step 7:Upon receiving the message from $SN_{j}$, $GWN_{j}$ firstly verifies if $T_{8}-T_{7}\leq ∆T$ holds. If this is not true then the session expires. Otherwise, $GWN_{j}$ calculates $R_{SN_{j}}^{*}=M_{8}⊕h\left(R_{GWN_{j}}\|\;R_{BS}\;\|\;T_{7}\right)$, $SK^{*}=h\left(R_{GWN_{j}}\;\|\;R_{U}\;\|\;R_{SN_{j}}^{*}\|R_{BS}\;\|\;ID_{i}\;\|\;M_{1}\right)$ and $M_{9}^{*}=h\left(SK^{*}\;\|\;R_{BS}\;\|\;R_{U}\;\|\;M_{2}\;\|\;T_{7}\right)$, then checks if $M_{9}^{*}{\;}_{=}^{?}\;M_{9}$ holds. If the condition is true then further is proceeded. Otherwise, the session is terminated.Step 8:The $GWN_{j}$ computes $M_{10}=h\left(R_{U}\;\|\;ID_{i}\right)⊕\left(R_{GWN_{j}}\|\;R_{SN_{j}}^{*}\|\;R_{BS}\right)$ and $M_{11}=\;h\left(R_{U}\;\|\;R_{SN_{j}}^{*}\|\;M_{2}\;\|\;SK\;\|\;T_{9}\right)$. Then, $GWN_{j}$ sends $\left\{\;M_{2},\;\;M_{10},\;M_{11},\;T_{9}\;\right\}$ to $U_{i}$.Step 9:Upon receiving the message from $GWN_{j}$, $U_{i}$ firstly verifies if $T_{8}-T_{7}\leq ∆T$ holds. If this does not true then session expires. Otherwise, $U_{i}$ computes $\left(R_{GWN_{j}}\|R_{SN_{j}}^{*}\|R_{BS}\right)$=$h\left(R_{U}\|ID_{i}\right)$⊕$M_{10}$, $SK^{*}$=$h\left(R_{GWN_{j}}\|R_{U}\|R_{SN_{j}}^{*}\|R_{BS}\|ID_{i}\|M_{1}\right)$, $h\left(A_{i}^{new}\;\|\;X\right)=M_{2}⊕h\left(h\left(A_{i}\|X\right)\;\|\;R_{BS}\right)$, $M_{11}^{*}=h\left(R_{U}\;\|\;R_{SN_{j}}^{*}\|\;M_{2}\;\|\;SK^{*}\|\;T_{9}\right)$ and verifies if $M_{11}^{*}{\;}_{=}^{?}\;M_{11}$ holds. If the condition is true then mutual authentication and session key agreement holds. Otherwise, the session is terminated.Step 10:The $U_{i}$ computes $B_{i}^{new}=h\left(A_{i}^{new}\;\|\;X\right)⊕h\left(ID_{i}\|\;RPW_{i}\right)$ and $D_{i}^{new}=\;h\left(A_{i}^{new}\;\|\;RPW_{i}\;\|\;ID_{i}\right)$. Then, $U_{i}$ replaces $A_{i},B_{i},D_{i}$ with $A_{i}^{new}$, $B_{i}^{new}$, $D_{i}^{new}$, respectively.

### 3.4. Password Updates or Change Phase

As shown in Figure 8, the following steps were executed to update or change user’s password.
Step 1:The user $U_{i}$ inserts his/her own smartcard into card reader and enters $ID_{i}$, $PW_{i}$ and imprints $F_{i}$ on sensor device. The card reader computes $Rep(F_{i}$, $P_{F})$ = $X_{F}^{*}$, $RPW_{i}^{*}=$
$h\left(PW_{i}\;\|\;X_{F}^{*}\right)$, $A_{i}^{*}=h(RPW_{i}^{*}\;\|\;ID_{i}\;\|\;X_{F}^{*})$, ${\left[h(A_{i}\;\|\;X)\right]}^{*}$ = $B_{i}⊕h\left(ID_{i}\;\|\;RPW_{i}^{*}\right)$ and $D_{i}^{*}=h\left(A_{i}\|\;RPW_{i}^{*}\;\|\;ID_{i}\right)$. Then, $U_{i}$ verifies if $D_{i}^{*}{\;}_{=}^{?}\;D_{i}$ holds. If condition is true then further is proceeded. Otherwise, the session is terminated.Step 2:The $U_{i}$ enters new password $PW_{i}^{new}$ and computes $RPW_{i}^{new}=h\left(PW_{i}^{new}\;\|\;X_{F}^{*}\right)$, $B_{i}^{new}=h\left(A_{i}\|X\right)⊕h\left(ID_{i}\|RPW_{i}^{new}\right)$ and $D_{i}^{new}=\;h\left(A_{i}\;\|\;RPW_{i}^{new}\;\|\;ID_{i}\right)$. Then, $\left\{B_{i},\;D_{i}\right\}$ are replaced with $\left\{B_{i}^{new},\;D_{i}^{new}\right\}$, respectively.

## 4. Security Analysis

### 4.1. Authentication Proof of the Proposed Scheme Using BAN Logic

This section validates session key agreement and mutual authentication of the proposed scheme using BAN (Burrows-Abadi-Needham) logic [31]. The BAN includes a set of rules to verify the message source, freshness, and trustworthiness of the scheme. Table 2 lists the notations and their respective abbreviations related to the BAN logic.

#### 4.1.1. Basic Rules of BAN Logic

Some rules or logical postulates used in the BAN logic are given as follows:**Rule 1. Message-meaning rule:**$\frac{P|\equiv P\overset{K}{↔}Q,P\;⨞\;{\left\{X\right\}}_{K}\;}{P|\equiv Q|\sim X}$If the entity P believes that the secret K is shared with Q and sees message X is encrypted using K, then P believes that Q once said X.**Rule 2. Jurisdiction rule:**$\frac{P|\equiv Q⟹X,\;P|\equiv Q|\equiv X\;}{P|\equiv X}$If the entity P believes that Q has jurisdiction over X and Q believes X, then P believes that X is true.**Rule 3. Nonce-verification rule:**$\;\frac{P|\equiv \#\left(X\right),\;\;P|\equiv Q|\sim X\;}{P|\equiv Q|\equiv X}$If the entity P believes that X is fresh and the entity Q once said X, then P believes that Q believes X.**Rule 4. Session key rule:**$\;\frac{P|\equiv \#\left(X\right),P|\equiv Q|\equiv X\;}{P|\equiv P\overset{K}{↔}Q}$If the entity P believes that X is fresh and Q believes X, then P believes the secret K that is shared between both entities P and Q.**Rule 5. Freshness-conjuncatenation rule:**$\;\frac{P|\equiv \#\left(X\right)}{P|\equiv \#\left(X,\;Y\right)}$If the entity P believes that X is fresh, then P believes the freshness of (X, Y).

#### 4.1.2. Goals

The proposed scheme needs to satisfy the following goals to ensure its security under BAN logic, using the above assumptions and postulates.
**Goal 1:**$BS\;|\equiv \;U_{i}\overset{SK}{↔}BS$**Goal 2:**$BS\;|\equiv U_{i}|\;\equiv U_{i}\overset{SK}{↔}BS$**Goal 3:**$GWN_{j}\;|\equiv BS\overset{SK}{↔}GWN_{j}$**Goal 4:**$GWN_{j}\;|\equiv BS\;|\equiv BS\overset{SK}{↔}GWN_{j}$**Goal 5:**$SN_{j}\;|\equiv GWN_{j}\overset{SK}{↔}SN_{j}$**Goal 6:**$SN_{j}\;|\equiv GWN_{j}\;|\equiv GWN_{j}\overset{SK}{↔}SN_{j}$**Goal 7:**$GWN_{j}\;|\equiv \;SN_{j}\overset{SK}{↔}GWN_{j}$**Goal 8:**$GWN_{j}\;|\equiv \;SN_{j}\;|\equiv \;SN_{j}\overset{SK}{↔}GWN_{j}$**Goal 9:**$U_{i}\;|\equiv \;GWN_{j}\overset{SK}{↔}U_{i}$**Goal 10:**$U_{i}\;|\equiv \;GWN_{j}\;|\equiv \;GWN_{j}\overset{SK}{↔}U_{i}$

#### 4.1.3. Idealized Form

Initially, the message of login, authentication, and key agreement scheme in the proposed scheme can be transformed into idealized form in the following manner.
**Message 1.**$\left(U_{i}\;BS\right)\;:A_{i}$, $DID_{i}$, $T_{1}$, $M_{1}$, $ID_{SN_{j}}$, $ID_{GWN_{j}}$: ${\langle R_{U}\rangle }_{h\left(A_{i}∥X\right)}$**Message 2.**$\left(BS\;GWN_{j}\right)\;:$$M_{1}$, $M_{2}$, $M_{3}$, $M_{4}$, $M_{5}$, $T_{3}$: ${\langle R_{U},\;R_{BS}\rangle }_{X_{BS-GWN_{j}}}$**Message 3.** ($GWN_{j}\;SN_{j})\;:$
$M_{1}$, $M_{2}$, $M_{5}$, $M_{6}$, $M_{7}$, $T_{3}$, $T_{5}$: ${\langle R_{U},\;R_{GWN_{j}}\rangle }_{R_{BS}}$**Message 4.** ($SN_{j}\;GWN_{j})\;:M_{1}$, $M_{2}$, $M_{8}$, $M_{9}$, $T_{7}$: ${\langle R_{SN_{j}}\rangle }_{R_{GWN_{j}}∥R_{BS}}$**Message 5.** ($GWN_{j}\;U_{i})\;:$
$M_{2}$, $M_{10}$, $M_{11}$, $T_{9}$: ${\langle R_{GWN_{j}},\;R_{SN_{j}},R_{BS}\rangle }_{h\left(R_{u}∥ID_{i}\right)}$

#### 4.1.4. Assumptions

The following initial assumptions have been established to prove the security of the proposed scheme using BAN logic.
***A*_1_:**$U_{i}$ |≡ #($R_{U}$, $R_{BS}$, $R_{GWN_{j}}$, $R_{SN_{j}}$)***A*_2_:**BS |≡ #($R_{U}$, $R_{BS}$)***A*_3_:**$GWN_{j}$ |≡ #($R_{U}$, $R_{BS}$, $R_{GWN_{j}}$, $R_{SN_{j}}$)***A*_4_:**$SN_{j}$ |≡ #($R_{U}$, $R_{BS}$, $R_{GWN_{j}}$, $R_{SN_{j}}$)***A*_5_:**BS |≡
BS
$\overset{\;\;\;h\left(A_{i}∥X\right)\;\;\;}{↔}$
$U_{i}$***A*_6_:**BS |≡
$U_{i}$
⟹
$R_{U}$***A*_7_:**$GWN_{j}$ |≡
$GWN_{j}$
$\overset{X_{BS-GWN_{j}}}{↔}$
BS***A*_8_:**$GWN_{j}$ |≡ BS⟹
$R_{BS}$***A*_9_:**$SN_{j}$ |≡
$SN_{j}$
$\overset{R_{BS}}{↔}$
$GWN_{j}$***A*_10_:**$SN_{j}$ |≡
$GWN_{j}⟹$
$R_{GWN_{j}}$***A*_11_:**$GWN_{j}$ |≡
$GWN_{j}$
$\overset{R_{GWN_{j}\;}∥\;R_{BS}}{↔}$
$SN_{j}$***A*_12_:**$GWN_{j}$ |≡
$SN_{j}$
⟹
$R_{SN_{j}}$***A*_13_:**$U_{i}$ |≡
$U_{i}$
$\overset{h(R_{U}∥ID_{i})}{↔}$
$GWN_{j}$***A*_14_:**$U_{i}$ |≡
$GWN_{j}$
⟹
$R_{GWN_{j}},\;R_{SN_{j}},R_{BS}$

#### 4.1.5. Verification

Verification shows the correctness of the proposed scheme confirmed by analyzing the idealized form using the above assumptions and the rules of the BAN logic.

By using **Message 1**:

***V*_1_:**BS ⨞ {$A_{i}$, $DID_{i}$, $T_{1}$, $M_{1}$, $ID_{SN_{j}}$, $ID_{GWN_{j}}$: ${\langle R_{U}\rangle }_{h\left(A_{i}∥X\right)}$}

From ***A*_5_**, ***V*_1_** and Rule 1:

***V*_2_:**BS |≡
$U_{i}$ |~$\;R_{U}$

From ***A*_2_**, ***V*_2_** and **Rule 3**:

***V*_3_:**BS |≡
$U_{i}$ |$\equiv \;R_{U}$

Then, from ***A*_6_**, ***V*_3_** and **Rule 2**:

***V*_4_:**BS |$\equiv \;R_{U}$

According to ***A*_2_**, ***V*_3_** and **Rule 4**:
***V*_5_:**BS |≡
BS
$\overset{\;\;\;sk\;\;\;}{↔}$
$U_{i}$
  ***Goal*_1_**

Further, using ***A*_2_**, ***V*_5_** and **Rule 3**:
***V*_6_:**BS |≡
$U_{i}$ |≡
BS
$\overset{\;\;\;sk\;\;\;}{↔}$
$U_{i}$
  ***Goal*_2_**

By using **Message 2**:

***V*_6_:**$GWN_{j}\;⨞$ {$M_{1}$, $M_{2}$, $M_{3}$, $M_{4}$, $M_{5}$, $T_{3}$: ${\langle R_{U},\;R_{BS}\rangle }_{X_{BS-GWN_{j}}}$}

From ***A*_7_**, ***V*_7_** and **Rule 1**:

***V*_8_:**$GWN_{j}$ |≡
BS |~$R_{BS}$

From ***A*_3_**, ***V*_8_** and **Rule 3**:

***V*_9_:**$GWN_{j}$ |≡
BS |≡
$R_{BS}$

Then, from ***A*_8_**, ***V*_9_** and **Rule 2**:

***V*_10_:**$GWN_{j}$ |≡
$R_{BS}$

According to ***A*_3_**, ***V*_9_** and **Rule 4**:
***V*_11_:**$GWN_{j}$ |≡
$GWN_{j}$
$\overset{\;\;\;sk\;\;\;}{↔}$
BS
  ***Goal*_3_**

Further, using ***A*_3_**, ***V*_11_** and **Rule 3**:
***V*_12_:**$GWN_{j}$ |≡
BS |≡
$GWN_{j}$
$\overset{\;\;\;sk\;\;\;}{↔}$
BS
  ***Goal*_4_**

By using **Message 3**:

***V*_13_:**$SN_{j}$ ⨞ {$M_{1}$, $M_{2}$, $M_{5}$, $M_{6}$, $M_{7}$, $T_{3}$, $T_{5}$: ${\langle R_{U},\;R_{GWN_{j}}\rangle }_{R_{BS}}$}

From ***A*_9_**, ***V*_13_** and **Rule 1**:

***V*_14_:**$SN_{j}$ |≡
$GWN_{j}$ |~ $R_{GWN_{j}}$

From ***A*_4_**, ***V*_14_** and **Rule 3**:

***V*_15_:**$SN_{j}$ |≡
$GWN_{j}$ |≡
$R_{GWN_{j}}$

Then, from ***A*_10_**, ***V*_15_** and **Rule 2**:

***V*_16_:**$SN_{j}$ |≡
$R_{GWN_{j}}$

According to ***A*_4_**, ***V*_15_** and **Rule 4**:
***V*_17_:**$SN_{j}$ |≡
$SN_{j}$
$\overset{\;\;\;sk\;\;\;}{↔}$
$GWN_{j}$
  ***Goal*_5_**

Further, using ***A*_4_**, ***V*_17_** and **Rule 3**:
***V*_18_:**$SN_{j}$ |≡
$GWN_{j}$ |≡
$SN_{j}$
$\overset{\;\;\;sk\;\;\;}{↔}$
$GWN_{j}$
  ***Goal*_6_**

By using **Message 4**:

***V*_19_:**$GWN_{j}$ ⨞ {$M_{1}$, $M_{2}$, $M_{8}$, $M_{9}$, $T_{7}$: ${\langle R_{SN_{j}}\rangle }_{R_{GWN_{j}}∥R_{BS}}$}

From ***A*_11_**, ***V*_19_** and **Rule 1**:

***V*_20_:**$GWN_{j}$ |≡
$SN_{j}$ |~ $R_{SN_{j}}$

From ***A*_3_**, ***V*_20_** and **Rule 3**:

***V*_21_:**$GWN_{j}$ |≡
$SN_{j}$ |≡
$R_{SN_{j}}$

Then, from ***A*_12_**, ***V*_21_** and **Rule 2**:

***V*_22_:**$GWN_{j}$ |≡
$R_{SN_{j}}$

According to ***A*_3_**, ***V*_21_** and **Rule 4**:
***V*_23_:**$GWN_{j}$ |≡
$GWN_{j}$
$\overset{\;\;\;sk\;\;\;}{↔}$
$SN_{j}$
  ***Goal*_7_**

Further, using ***A*_4_**, ***V*_17_** and **Rule 3**:
***V*_24_:**$GWN_{j}$ |≡
$SN_{j}$ |≡
$GWN_{j}$
$\overset{\;\;\;sk\;\;\;}{↔}$
$SN_{j}$
  ***Goal*_8_**

By using **Message 5**:

***V*_25_:**$U_{i}$ ⨞ {$M_{2}$, $M_{10}$, $M_{11}$, $T_{9}$: ${\langle R_{GWN_{j}},\;R_{SN_{j}},R_{BS}\rangle }_{h\left(R_{u}∥ID_{i}\right)}$}

From ***A*_13_**, ***V*_25_** and **Rule 1**:

***V*_26_:**$U_{i}$ |≡
$GWN_{j}$ |~ $R_{GWN_{j}},\;R_{SN_{j}},R_{BS}$

From ***A*_1_** and **Rule 5**:

***V*_27_:**$U_{i}$ |≡ #($R_{GWN_{j}},\;R_{SN_{j}},R_{BS}$)

Then, from ***V*_26_**, ***V*_27_** and **Rule 3**:

***V*_28_:**$U_{i}$ |≡
$GWN_{j}$ |≡
$R_{GWN_{j}},\;R_{SN_{j}},R_{BS}$

Moreover, from ***A*_14_**, ***V*_28_** and **Rule 2**:

***V*_29_:**$U_{i}$ |≡
$R_{GWN_{j}},\;R_{SN_{j}},R_{BS}$

According to ***V*_27_**, ***V*_28_** and **Rule 4**:
***V*_30_:**$U_{i}$ |≡
$U_{i}\overset{\;\;\;sk\;\;\;}{↔}$
$GWN_{j}$
  ***Goal*_9_**

Then, using ***V*_27_**, ***V*_30_** and **Rule 3**:
***V*_31_:**$U_{i}$ |≡
$GWN_{j}$ |≡
$U_{i}\overset{\;\;\;sk\;\;\;}{↔}$
$GWN_{j}$
  ***Goal*_10_**

### 4.2. Informal Security Analysis

This section presents informal security analysis of the proposed scheme. Table 2 summarizes security analysis comparisons between Ali et al.’s scheme [23] and the proposed scheme.

#### 4.2.1. User Anonymity

When base station BS gets a request message $\left\{A_{i},\;\;DID_{i},\;\;T_{1},\;\;M_{1},\;\;ID_{SN_{j}},\;\;ID_{GWN_{j}}\right\}$ from user $U_{i}$, base station BS checks whether the request message comes from a legitimate user $U_{i}$ by calculating $\left(ID_{i}\|R_{U}\right)=DID_{i}⊕h(h(A_{i}\|X)\|T_{1}$), where X is BS’s secret key that is known only by BS. BS checks whether $h\left(R_{U}\;\|\;ID_{i}\;\|\;T_{1}\;\|\;h(A_{i}\|X\right))$ equals $M_{1}$. If it holds, BS confirms that the request message is coming from a legitimate $U_{i}$.

Assume an adversary tries to get the real identity $ID_{i}$ of a legitimate user $U_{i}$ from an existing message $\left\{A_{i},\;DID_{i},\;T_{1},\;M_{1},\;ID_{SN_{j}},\;ID_{GWN_{j}}\right\}$ that can be obtained from public channel. In order to successfully get the real $ID_{i}$, the adversary needs to calculate $DID_{i}⊕h(h(A_{i}\|X)\|T_{1}$). However, X is only known by the legitimate BS and is also protected by the hash operation that makes X is computationally infeasible to calculate. Without knowledge of X, the adversary cannot derive the real $ID_{i}$. Therefore, the proposed scheme provides user anonymity.

#### 4.2.2. User Traceability

In the proposed scheme, the real identity $ID_{i}$ of a user is protected by using dynamic pseudonym identity $DID_{i}$, where $DID_{i}=\left(ID_{i}\|R_{U}\right)⊕h(h(A_{i}\|X)\|T_{1}$). $R_{U}$ is random and $T_{1}$ is timestamp that newly generated for each transaction, means $DID_{i}$ is dynamic for every transaction. Moreover, different values of $A_{i},\;\;DID_{i},\;\;T_{1},\;\;M_{1}$ for each transaction prevents adversaries to identify a transaction belonging to whom or related with any specific user. Therefore, the proposed scheme provides protection to user traceability.

#### 4.2.3. Three-Factor Security

To provide protection in the login phase, the proposed scheme uses three-factor security which means only a user with the correct password, correct biometric characteristics, and correct smart card is allowed to login to the remote server [32].

Assume an adversary has any two factors of security which are password and smartcard, or smartcard and biometric, or password and smartcard. When he/she tries to login into the system, the proposed scheme will check whether $h\left(A_{i}\|\;RPW_{i}^{*}\;\|\;ID_{i}\right)$ equals with $D_{i}$, where $A_{i}$ = $h(RPW_{i}^{*}\;\|\;ID_{i}\;\|\;X_{F}^{*})$ and $RPW_{i}^{*}$ = $h\left(PW_{i}\;\|\;X_{F}^{*}\right)$. Based on this checking, the system always completely checks three factor security first before allowed any request to successfully login into the system. This process means an adversary who has only two factors of security does not have a chance to enter into the system. Therefore, the proposed scheme provides three-factor security.

#### 4.2.4. Session Key Security

In an authentication and key agreement phase, the session key SK must be made and known only by legal participants. In the proposed scheme, SK is computed by using random numbers from each legal participant that freshly generated in each session. Furthermore, SK also depends on $ID_{i}$ and $M_{1}$, where $M_{1}=h\left(R_{U}\;\|\;ID_{i}\;\|\;T_{1}\|\;h(A_{i}\|X\right))$ and it was protected by $h(A_{i}\|X)$ that is only known by $U_{i}$ and BS. It is also computationally infeasible to calculate the session key $SK=h\left(R_{GWN_{j}}\;\|\;R_{U}\;\|\;R_{SN_{j}}\;\|\;R_{BS}^{*}\;\|\;ID_{i}\;\|\;M_{1}\right)$ due to the characteristics of the hash operation. Therefore, the proposed scheme withstands session key computation attack.

#### 4.2.5. Perfect Forward Secrecy Attack

The proposed scheme ensures the secrecy of previous session keys even if the master secret key of the server or shared secret key between legal participants are compromised.

In the proposed scheme, the session key is not related to the master secret key X that belongs to the base station BS. Also, the session key is not related with any shared secret key that exists between legal participants, such as the shared key between BS and gateway node $X_{BS-GWN_{j}}$ or the shared key between BS and sensor node $RI_{j}$. Instead, the session key is built from each random number that is freshly generated by every legal participant from each session. Therefore, the proposed scheme provides perfect forward secrecy.

#### 4.2.6. Sensor Node Impersonation Attack

Assume an adversary tries to impersonate a sensor node by sending a request message $\left\{\;M_{1},\;\;M_{2},\;\;M_{8},\;\;M_{9},\;T_{7}\right\}$ to a gateway node $GWN_{j}$. Upon receiving the request message, to verify if the request message comes from a legitimate sensor node $SN_{j}$ or not, $GWN_{j}$ computes $M_{9}^{*}=h\left(SK^{*}\;\|\;R_{BS}\;\|\;R_{U}\;\|\;M_{2}\;\|\;T_{7}\right)$ and checks if $M_{9}^{*}$ equals $M_{9}$ or not.

In the proposed scheme, in order to compute verifiable $M_{9}$, both $SN_{j}$ and $GWN_{j}$ need to obtain $R_{BS}^{*}$, where $R_{BS}$ is a random nonce belongs to the base station BS. As shown in authentication and session key agreement phase, in order to obtain $R_{BS}$, both $SN_{j}$ and $GWN_{j}$ need to use their own shared secret key with the BS, where $GWN_{j}$ uses its shared secret key $X_{BS-GWN_{j}}$ and $SN_{j}$ uses its shared secret key $RI_{j}$. Without shared secret key, an adversary will not be able to obtain $R_{BS}$. Without the right $R_{BS}$, $M_{9}$ will never be successfully verified by $GWN_{j}$. By verifying the $M_{9}$, $GWN_{j}$ will immediately detect that the request message is coming from legal $SN_{j}$ or not. Therefore, the proposed scheme withstands sensor node impersonation attack.

#### 4.2.7. Gateway Node Impersonation Attack

Gateway node impersonation attack occurs when an adversary acts as a legitimate gateway node $GWN_{j}$ by sending a request message to user $U_{i}$ or sensor node $SN_{j}$ and that request message is successfully authenticated as a legitimate $GWN_{j}$ by $U_{i}$ or $SN_{j}$.

Assume an adversary tries to impersonate $GWN_{j}$ by sending a request message $\left\{\;M_{1},M_{2},M_{5},M_{6},\;M_{7},\;T_{3},T_{5}\right\}$ to $SN_{j}$ or $\left\{\;M_{2},M_{10},M_{11},T_{9}\;\right\}$ to $U_{i}$, where $M_{6}=\left(R_{U}\;\|\;R_{GWN_{j}}\;\|\;ID_{i}\right)⊕h\left(R_{BS}\;\|\;M_{5}\;\|\;T_{5}\right)$, $M_{7}=h\left(M_{1}\;\|\;M_{2}\;\|\;R_{BS}\;\|R_{GWN_{j}}\|\;ID_{i}\;\|\;R_{U}\right)$, $M_{10}=h\left(R_{U}\;\|\;ID_{i}\right)⊕\left(R_{GWN_{j}}\;\|\;R_{SN_{j}}^{*}\;\|\;R_{BS}\right)$ and $M_{11}=\;h\left(R_{U}\;\|\;R_{SN_{j}}^{*}\|\;M_{2}\;\|\;SK\;\|\;T_{9}\right)$. To compute $M_{6}$, $M_{7}$, $M_{10}$ and $M_{11}$, the adversary needs to obtain $R_{U}$, $R_{BS}$ and $ID_{i}$ using shared key $X_{BS-GWN_{j}}$, where $\left(R_{U}\;\|\;R_{BS}\;\|\;ID_{i}\right)=M_{3}⊕h\left(X_{BS-GWN_{j}}\;\|\;T_{3}\right)$. However, without the knowledge of $X_{BS-GWN_{j}}$, it is computationally infeasible to calculate these parameters due to the characteristics of the hash operation. Without the right parameters, $U_{i}$ and $SN_{j}$ will immediately recognize if the request is not coming from a legitimate $GWN_{j}$. Therefore, the proposed scheme withstands a gateway node impersonation attack.

#### 4.2.8. User/Agriculture Impersonation Attack

A user impersonation attacks occur when an adversary acts as a legitimate user and is successfully authenticated by the base station BS.

Assume an adversary tries to impersonate a legitimate user $U_{i}$ by sending a request message $\left\{A_{i},DID_{i},T_{1},M_{1},ID_{SN_{j}},ID_{GWN_{j}}\right\}$ to BS, where $DID_{i}=\left(ID_{i}\;\|\;R_{U}\right)⊕h(h(A_{i}\|X)\;\|\;T_{1})$, $M_{1}=h\left(R_{U}\;\|\;ID_{i}\;\|\;T_{1}\|\;h(A_{i}\|X\right))$, ${\left[h(A_{i}\|X)\right]}^{*}=B_{i}⊕h\left(ID_{i}\;\|\;RPW_{i}^{*}\right)$ and $RPW_{i}^{*}=h\left(PW_{i}\;\|\;X_{F}^{*}\right)$. However, it is impossible for an adversary to calculate ${\left[h(A_{i}\;\|\;X)\right]}^{*}$ due to biometrics, unknown user identity $ID_{i}$ and user password $PW_{i}$.

Upon receiving the request message from $U_{i}$, BS will immediately recognize that the request message is coming from a legitimate user or not by checking whether $h\left(R_{U}\;\|\;ID_{i}\;\|\;T_{1}\;\|\;h(A_{i}\|X\right))$ equals $M_{1}$ or not. Therefore, the proposed scheme withstands user/agriculture impersonation attack.

#### 4.2.9. Offline Password Guessing Attack

An off-line password guessing attack occurs when a smart card is lost or stolen and the adversary tries to guess the password to log into the system. Let us assume an adversary obtains information within the smart card by using channel side attacks and successfully obtains $\left\{A_{i},B_{i},\;D_{i},h\left(.\right),\;P_{F},\;Gen\left(.\right)\right\}$, where $B_{i}=h\left(A_{i}\|X\right)⊕h\left(ID_{i}\|RPW_{i}\right)$, $D_{i}=\;h\left(A_{i}\;\|\;RPW_{i}\;\|\;ID_{i}\right)$, and $RPW_{i}=h\left(PW_{i}\;\|\;X_{F}\right)$. To guess the password through the parameter that are stored inside the smart card, the adversary needs to invert the value of $B_{i}$ or $D_{i}$. However, inverting the values of $B_{i}$ or $D_{i}$ is computationally infeasible due to the characteristics of the hash operation. Neither $ID_{i}$. or $PW_{i}$ are ever directly revealed or exposed and an adversary for sure cannot guess or change the password. Therefore, the proposed scheme withstands an offline password guessing attack.

#### 4.2.10. Replay Attack

An off-line password guessing attack occurs when a smart card is lost or stolen and the adversary replay attack happens when an adversary tries to retransmit previous request message as a new transaction request and it has successfully been accepted as a new legitimate request by other legal participants.

Assume an adversary tries to replay existing messages as a new transaction request. However, any message contains a timestamp $T_{1}$, $T_{3}$, $T_{5}$, $T_{7}$, or $T_{9}$. Other legitimate participants will immediately identify the replay attack when they check the freshness of $T_{1}$, $T_{3}$, $T_{5}$, $T_{7}$, and $T_{9}$. Therefore, the proposed scheme withstands replay attack.

#### 4.2.11. Insider Attack

An insider attack happens when a malicious legal participant successfully captures key values of others, such as a shared key, and then uses that key to launch some security violations or attacks. The proposed scheme ensures that the key shared between participants is known only by the right participant and will never be leaked to other irrelevant participants. 

Assume a malicious legal participant tries to obtain a shared secret key that belongs to another legal participant. In the proposed scheme, there are three shared secret key which are $X_{\mathrm{BS}-{\mathrm{GWN}}_{j}}$, ${\mathrm{RI}}_{j}$ and $h(A_{i}\|X)$. All of them are generated by base station BS and contains BS secret key X. Since the secret key of BS is only known by BS and never revealed to others, and since the shared key between participants is never revealed to other irrelevant participants too, the proposed scheme is safe from shared secret key leakage. Therefore, the proposed scheme withstands insider attack.

## 5. Performance and Functionality Comparisons

This section analyzes and compares Ali et al.’s scheme with the proposed scheme. Security functionality comparisons and performance comparisons in login, authentication, and key agreement phase are presented as follows.

### 5.1. Security Functionality Comparisons

Table 3 shows comparisons between the proposed scheme and Ali et al.’s scheme in terms of functionality in security. It shows that the proposed scheme enables provision of more security functionality where there are lacks in Ali et al.’s scheme. Detailed explanations about how Ali et al.’s scheme suffers from those attacks was already described in Section 2.3.

### 5.2. Performance Comparisons

As WSNs has limited power capacity, the computation cost for login, authentication, and key-agreement scheme must be made as minimal as possible. Table 4 shows the comparison of login, authentication, and key agreement phases between the proposed scheme and Ali et al.’s scheme in terms of performance. Table 5 shows hardware/software specifications and used algorithms in our simulation environment. The proposed scheme involves a user $U_{i}$, a base station BS, a gateway node $GWN_{j}$, and a sensor node $SN_{j}$. $T_{H}$ denotes the execution time of hash operation and $T_{s}$ donates the execution time of symmetric encryption/decryption.

Ali et al.’s scheme [23] requires 19 hash function and nine symmetric en/decryption operations. Although the proposed scheme requires more hash function operations, it does not require nine symmetric en/decryption operations. Therefore, the proposed scheme provides better efficiency compared with the previous scheme.

## 6. Conclusions

This paper reviewed Ali et al.’s scheme and demonstrated that it cannot provide user anonymity, user traceability, session key security, and is insecure against insider attacks, perfect forward secrecy attacks, and sensor node impersonation attacks. Moreover, after a user successfully updates his/her password, Ali et al.’s scheme will immediately suffer from Denial of Service as a Service (DoSaaS) in its authentication and key agreement scheme. The proposed scheme eliminated those security weaknesses by proposed four new phases of six existing phases. To promote efficiency, the proposed scheme eliminates symmetric encryption–decryption computation that was used in the previous scheme and only utilizes hash operation in its computation. The proposed scheme not only eliminates weaknesses in Ali et al.’s scheme, but is also 80 times more efficient compares with Ali et al.’s scheme. The efficiency, security and functionalities showed in the proposed scheme overcomes Ali et al.’s scheme. Therefore, the proposed scheme is more suitable for agriculture monitoring using WSNs.

## Figures and Tables

**Figure 1 sensors-19-01146-f001:**
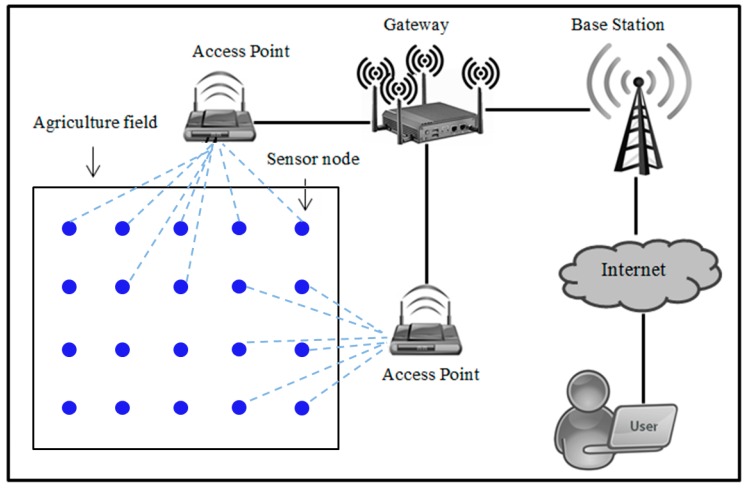
Agriculture monitoring system model of the proposed scheme.

**Figure 2 sensors-19-01146-f002:**
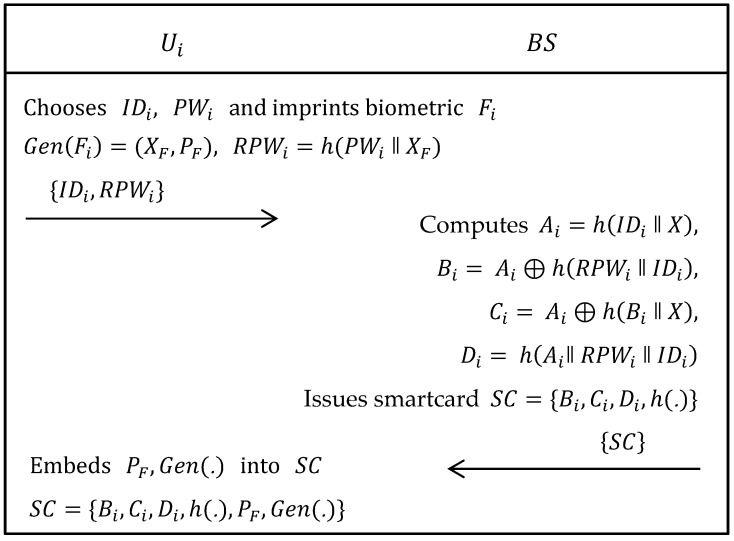
Registration phase of Ali et al.’s scheme.

**Figure 3 sensors-19-01146-f003:**
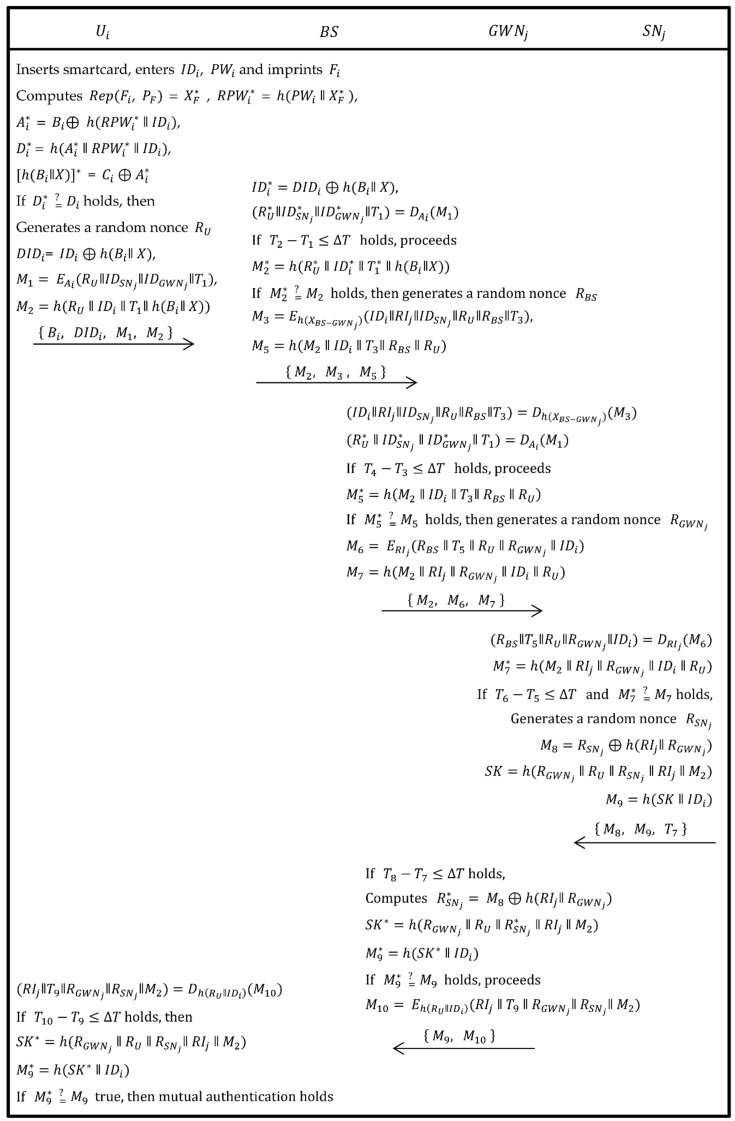
Login, authentication and key agreement phase of Ali et al.’s scheme.

**Figure 4 sensors-19-01146-f004:**
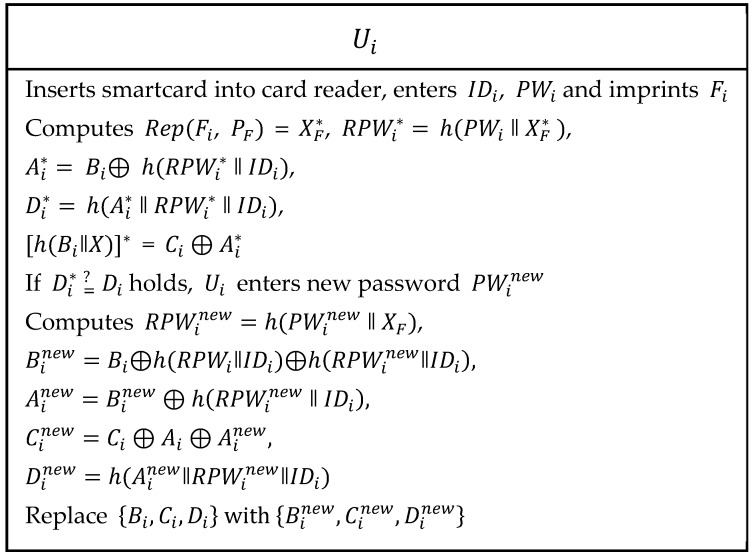
Password update phase of Ali et al.’s scheme.

**Figure 5 sensors-19-01146-f005:**
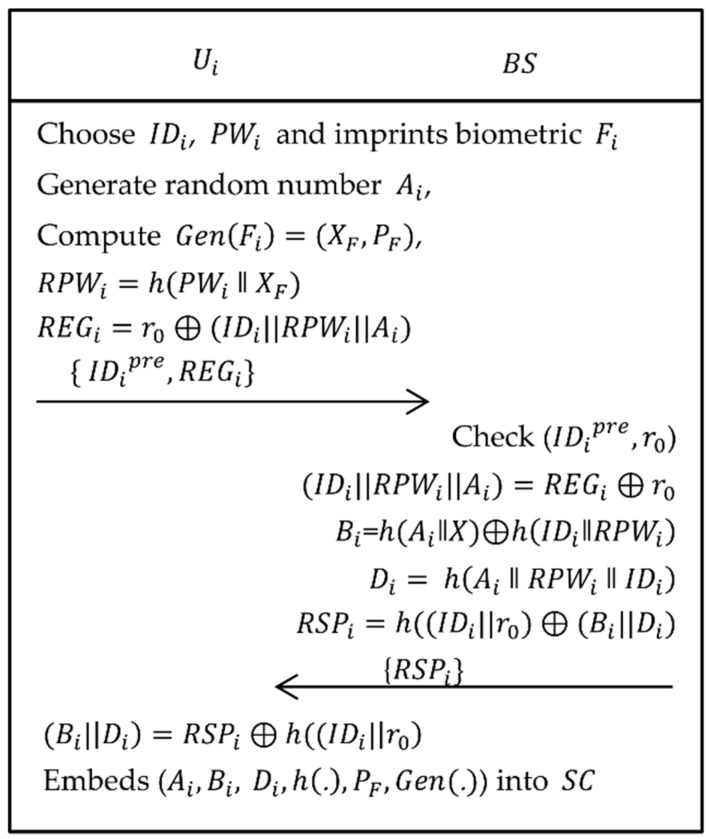
User registration phase of the proposed scheme.

**Figure 6 sensors-19-01146-f006:**
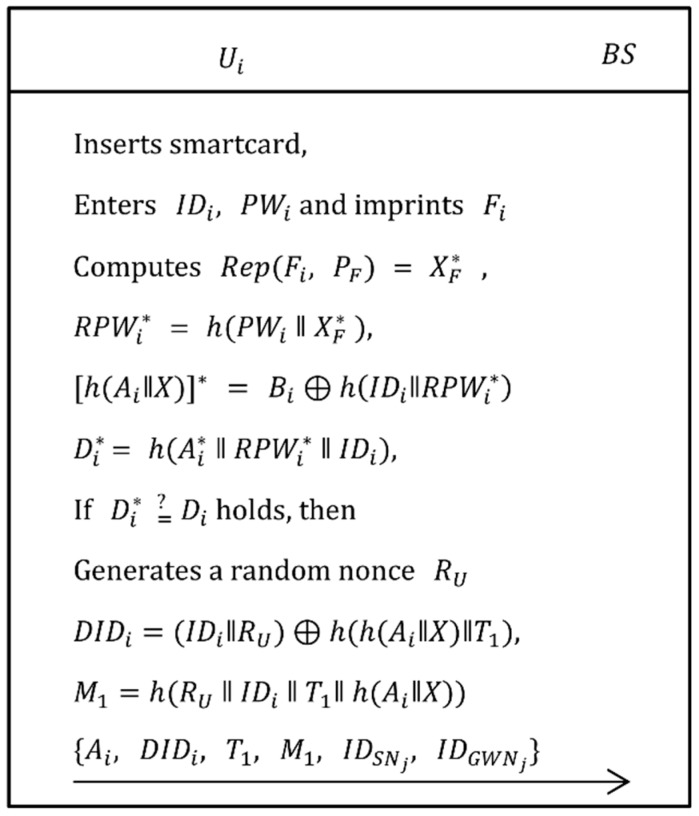
Login phase of the proposed scheme.

**Figure 7 sensors-19-01146-f007:**
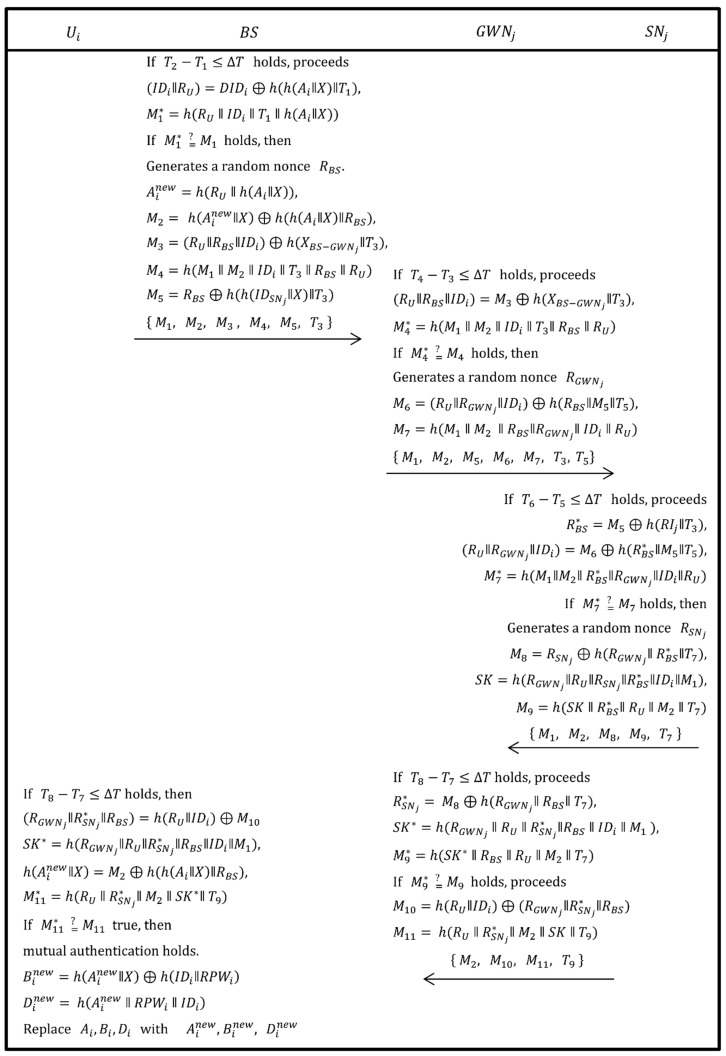
Authentication and key agreement phase of the proposed scheme.

**Figure 8 sensors-19-01146-f008:**
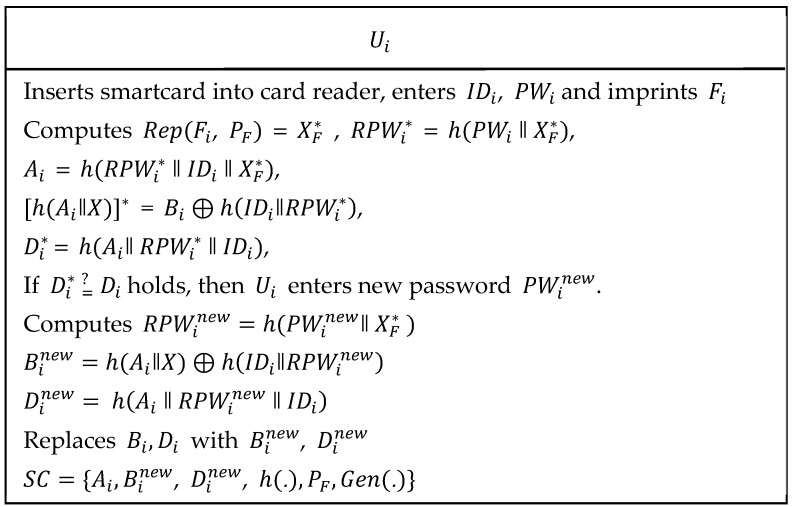
Password update phase of the proposed scheme.

**Table 1 sensors-19-01146-t001:** Notations of the proposed scheme.

Symbol	Description
$U_{i}$	User/agriculture professional
BS	Base station
$GWN_{j}$	Gateway node
$SN_{j}$	Sensor node
$ID_{i}$, $ID_{GWN_{j}}$, $ID_{SN_{j}}$	Identity of $U_{i}$, $GWN_{j}$, $SN_{j}$, respectively
$PW_{i}$	Password of $U_{i}$
$F_{i}$	Biometric of $U_{i}$
$A_{i}$	Shared key between BS and $U_{i}$
X	Secret key of BS
$X_{BS-GWN_{j}}$	Secret key shared between BS and $GWN_{j}$
$RI_{j}$	Secret key shared between BS and $SN_{j}$
$R_{U}$, $R_{BS}$, $R_{GWN_{j}}$, $R_{SN_{j}}$	Random nonce of $U_{i}$, $GWN_{j}$, $SN_{j}$, respectively
SK	Session key
$T_{i}$	Timestamp of *i*
$∆T_{i}$	Time differences between $T_{i}$
$E_{key}$, $D_{key}$	Encryption, Decryption using shared *key*, respectively
Gen(.)	Generate function of fuzzy extractor
Rep(.)	Reproduce function of fuzzy extractor
h(.)	Hash function
⊕	Exclusive OR operation
||	Concatenation operation

**Table 2 sensors-19-01146-t002:** BAN (Burrows-Abadi-Needham) logic notations and respective abbreviations.

Notation	Abbreviation
P |≡ X	The entity P believes the statement X
P⟹X	P has jurisdiction on the statement X
P |~ X	P once said X
P ⨞ X	P sees X
${\left\{X\right\}}_{K}$	Formula X is encrypted under the key K
$P\overset{K}{↔}Q$	P and Q communicate via shared key K
P→Q :m	P sends the message m and Q receives it
#X	The message #X is freshly generated

**Table 3 sensors-19-01146-t003:** Functionality comparisons.

Attributes	Ali et al. Scheme	Proposed Scheme
User anonymity	N	Y
User traceability	N	Y
Three-factor security	Y	Y
Session key security	N	Y
Perfect forward secrecy attack	N	Y
Sensor node impersonation attack	N	Y
Gateway node impersonation attack	Y	Y
User/agriculture impersonation attack	Y	Y
Offline password guessing attack	Y	Y
Replay attack	Y	Y
Insider attack	N	Y
Denial of Service as a Service	N	Y

**Table 4 sensors-19-01146-t004:** Performance comparisons.

	$U_{i}$	BS	$GWN_{j}$	$SN_{j}$	Total
Ali et al.’s scheme	$7T_{H}+2T_{S}$	$3T_{H}+2T_{S}$	$5T_{H}+4T_{S}$	$4T_{H}+1T_{S}$	$19T_{H}+9T_{S}$
Response time	0.00836 s	0.00832 s	0.01044 s	0.00514 s	0.03512 s
Proposed scheme	$14T_{H}$	$8T_{H}$	$9T_{H}$	$6T_{H}$	$37T_{H}$
Response time	000280 s	0.00176 s	0.00126 s	0.00144 s	0.00740 s

**Table 5 sensors-19-01146-t005:** Simulation environment.

Hardware/Software Specification		
$U_{i}$	Mainboard	ASUSTeK Computer INC. CM5571
	CPU	Intel Core 2 Quad Q8300 @ 2.50 GHz 2.50 GHz
	Memory	4.00 GB Dual-Channel DDR3 @ 533 MHz
	OS	Windows 7 64-bit SP1
BS	Mainboard	ASUSTeK Computer INC. CM5571
	CPU	Intel Core 2 Quad Q8300 @ 2.50 GHz 2.50 GHz
	Memory	4.00 GB Dual-Channel DDR3 @ 533 MHz
	OS	Windows 7 64-bit SP1
GWN	Mainboard	IBM 46W9191
	CPU	Intel Xeon E3 1231 v3 @ 3.40 GHz 3.40 GHz
	Memory	8.00 GB Dual-Channel DDR3 @ 800 MHz
	OS	Windows Server 2008 R2 Standard 64-bit SP1
$SN_{j}$	Mainboard	ASUSTeK Computer INC. UX303LN
	CPU	Intel Core i3/i5/i7 4xxx @ 1.70 GHz
	Memory	4.00 GB Single-Channel DDR3 @ 798 MHz
	OS	Windows 8.1 64-bit
Used Programming Language and Algorithms C/C++ Hash function: SHA-1		

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
