# Peer review of "An Enhanced Lightweight Dynamic Pseudonym Identity Based Authentication and Key Agreement Scheme Using Wireless Sensor Networks for Agriculture Monitoring"

_sensors, 2019, doi:10.3390/s19051146_

Reviewer 1 Report

This paper firstly analyzed an authentication and key agreement scheme for wireless sensor network (WSN) based agriculture monitoring, which is presented by Ali et al. The authors discovered that Ali et al.s scheme has many flaws and vulnerable to many attacks (including the insider). In order to enhance the security of Ali et al’.s scheme, secondly, the authors proposed a lightweight dynamic identity based user authentication scheme for WSN based agriculture monitoring.

The paper is well organized and well written. Few comments:

(i)             Recently, many of user authentication schemes have been studied and proposed in generic WSN. How these two schemes (i.e., Ali et al and the proposed scheme) are different than others. Does agriculture application requires different security requirements, which cannot be fulfilled by the generic proposed schemes?

(ii)            In my opinion, considering the insider attack – the scheme proposed by Ali et al.s is indeed vulnerable to sensor node impersonation attack, perfect forward secrecy, and violation of session key security and user anonymity.

(iii)            In the proposed scheme (page 9, Section IV), many assumptions are not clear, e.g., (i) how Ai is shared between the user and the base station (BS), at first place. (ii) Who is the system administrator (i.e., BS or some another entity) and how the keys are being distributed to the entities. Not enough clear.

(iv)          In the proposed scheme, the message M5 (refer to authentication and key agreement phase: Step 2) is computed via RBS ⊕ h(h(IDSNi||X)||T3) at the BS. Now, using M5, the SN derives RBS * = M5 ⊕ h(RJi ||T3), refer to authentication and key agreement phase: Step 5. I am wondering how RBS will be derived from M5, since these two expressions are different?

(v)           Regarding the security analysis, as the BAN logic itself has many inherent issues, this reviewer would like to recommend to use other software-based tools, e.g., Tamarin prover tool, or Scyther, for the formal security verification. 

(vi) In the informal analysis – what kind of attack model is being used, not enough clear. More importantly, how the new proposed scheme is secure against the insider attack, the authors should provide more justifications.

Author Response

This paper firstly analyzed an authentication and key agreement scheme for wireless sensor network (WSN) based agriculture monitoring, which is presented by Ali et al. The authors discovered that Ali et al.s scheme has many flaws and vulnerable to many attacks (including the insider). In order to enhance the security of Ali et al’.s scheme, secondly, the authors proposed a lightweight dynamic identity based user authentication scheme for WSN based agriculture monitoring.

The paper is well organized and well written. Few comments:

(i)   Recently, many of user authentication schemes have been studied and proposed in generic WSN. How these two schemes (i.e., Ali et al and the proposed scheme) are different than others. Does agriculture application requires different security requirements, which cannot be fulfilled by the generic proposed schemes? 

Answer: We thank the reviewer for this constructive suggestion. When we see how importance the agriculture for everyone, we know that we need to give a study that focus on it. Other general schemes for WSNs may not necessarily accordance with agriculture monitoring necessities. For example, even for the organization structures such as entities (user, sensor nodes, gateway nodes, cluster head, etc.) can be different with the general schemes because of its different designation purposes. Ali, et al. are the first authors whose can see these importance and then designed a special scheme for agriculture monitoring using WSNs. Unfortunately, their scheme suffered from major security weaknesses. Our study mentioned their weaknesses and proposed an improved scheme which is more secure and more efficient a lot. The related descriptions are shown on Pages 1-2, Sec. 1, Paragraphs 1 and 2.

(ii)  In my opinion, considering the insider attack – the scheme proposed by Ali et al.s is indeed vulnerable to sensor node impersonation attack, perfect forward secrecy, and violation of session key security and user anonymity.

Answer: We thank the reviewer for this constructive suggestion.

(iii)        In the proposed scheme (page 9, Section IV), many assumptions are not clear, e.g., (i) how Ai is shared between the user and the base station (BS), at first place. (ii) Who is the system administrator (i.e., BS or some another entity) and how the keys are being distributed to the entities. Not enough clear.

(iii.i) “how Ai is shared between the user and the base station (BS), at first place.”

Answer: We thank the reviewer for this constructive suggestion. There are 2 parts where Ai is shared with BS.

·       The first part, Ai is shared with BS in user registration phase (as shown in Figure 5) and Section 4.1, Step 1, through this message: {IDi, RPWi, Ai}. However, in our Section 4.1, Step 1, we just found that we did not put Ai into this statement: “Now, Ui sends {IDi, RPWi} to BS via trustworthy channel.”. We added Ai to {IDi, RPWi}. Now, in Section 4.1, Step 1 become “Now, Ui sends {IDi, RPWi, Ai} to BS via trustworthy channel.”.

·       The second part, Ai is shared with BS during login phase through this message: {Ai, DIDi, T1, Mi, IDSNj, IDGWNj}, as shown in Figure 6 and Section 4.2, Step 2.

BS never store Ai in its database. Ai always be declared openly during the login phase. The function of Ai is to make sure that Ui holds a legal shared key with BS, where the shared key between them is h(Ai||X).

(iii.ii) “Who is the system administrator and how the keys are being distributed to the entities.”

Answer: We thank the reviewer for this constructive suggestion.

·       The system administrator is the base station (BS). To make it clear, we added some information to make it clearer for reader.

·       Which part about the keys that are not clearly distributed?

(iv)         In the proposed scheme, the message M5 (refer to authentication and key agreement phase: Step 2) is computed via RBS ⊕ h(h(IDSNi||X)||T3) at the BS. Now, using M5, the SN derives RBS * = M5 ⊕ h(RJi ||T3), refer to authentication and key agreement phase: Step 5. I am wondering how RBS will be derived from M5, since these two expressions are different?

Answer: We thank the reviewer for this constructive suggestion.

·       First, we need to see the system setup phase in section 2.1, where RIj is a shared key between BS and SN, and RIj=h(IDSNj||X).

·       Now, we go to section 4.3, step 2, where M5 = RBS Å h(h(IDSNj||X)||T3) and section 4.3, step 5, where RBS* = M5 Å h(RIj ||T3).

·       Since RIj=h(IDSNj||X), then RBS* = M5 Å h(RIj ||T3) equals with RBS* = M5 Å h(h(IDSNj||X)||T3).

Now, we can clearly see that SN can derive RBS from M5. The descriptions are shown on Page 4, Sec. 2.1.

(v)   Regarding the security analysis, as the BAN logic itself has many inherent issues, this reviewer would like to recommend to use other software-based tools, e.g., Tamarin prover tool, or Scyther, for the formal security verification.

Answer: We thank the reviewer for this constructive suggestion. Indeed, the BAN logic cannot completely consider security requirements. We thus try to adopt the BAN logic analysis in Sec 4.1 and heuristic security analyses in Sec 4.2 to have complete security requirements. We leave this part for future work, as we need more time to apply such software-based tools.

(vi) In the informal analysis – what kind of attack model is being used, not enough clear. More importantly, how the new proposed scheme is secure against the insider attack, the authors should provide more justifications.

Answer: We thank the reviewer for this constructive suggestion. We have revised the descriptions according to the reviewer’s comments. The new proposed scheme has already mentioned in Section 3 about how the proposed scheme is secure against the insider attack and by then eliminates the sensor node impersonation attack, perfect forward secrecy and violation of user anonymity (lines 268-271 in Section 3). Additionally, we also provide more justifications about how the new proposed scheme is secure against the insider attack in Section 4.2.11.

Reviewer 2 Report

The paper proposes a lightweight authentication and key agreement scheme for wireless sensor networks. The authors improve a scheme already proposed by Ali et al. which they claim to be vulnerable to various attacks. Moreover, the authors demonstrate that their method is more lightweight than the one of Ali et al.

The paper seems sound but some doubts remain:

1) first of all: what is the connection with agriculture? why is this method propose donly for agriculture monitoring? It is a quite general scheme for wireless sensor neworks...please explain better or remove the references to the agriculture. Which are the security issues in the agriculture scenario?

2) literature review is too short, and you cannot group references like this [9-22]: more than 10 references grouped together with no description of their content. Extend the comparison to other key agreement and authentication schemes in peer-to-peer, sensor and distributed networks. I would suggest these references for example:

- https://ieeexplore.ieee.org/abstract/document/5306839

- https://ieeexplore.ieee.org/abstract/document/6364948

- https://ieeexplore.ieee.org/document/7057853

3) references 24-26 are not described

4) lines 95-100: you do not explain why anonymity is important in an agriculture scenario.

 5) title of section 2 is not appropiate....please select another one.. moreover your paper is only based on an imporvement on Ali et al.' s scheme...it seems a limitation to an incremental study

6) section 2.1: you do not say what is X

7)section 2.2, step 1: whose are the secret and public keys in the first line of page 4? the user's? you speak about a trustworthy channel: what is that? who guarantees that it is trustworthy?

8) what is the advantage of separating BS and GWN? you need more messages and a further entity

9) lines 134-136: what about the Sd , you name it and then it is not considered anymore

10) you do not make a comparison between your proposal and Ali et al.'s in terms of number of exchanged messages, are they exactly the same?

11) with which hardware or testbed have you measured the timing in lines 497-498?

12) Table 3: which phases are concerned in this time computation? All phases? compirsing also adding of new nodes and change of passwords?

13) have you thought about the security of using only hash operations against a replay attack?

Some English typos remain:

- line 24: have -> has

- line 32: plays AN important role...

- line 40: When -> While

- line 41: varying -> vary

- line 42: remains -> remain

- line 48: the people: remove THE

- line 53: integratING

- line 58: improved -> improve

- line 92: which WERE used

- line 64: the data -> data

- line 66: as A basis

- line 70-71: since THEY PLAY AN important role ... systemS and ARE very important (data is plural)

- line 72: data ARE exchanged

- line 144: belongING

- line 145: which user -> a certain user

- line 148: easily -> easy to

- line 151: happened -> happens..... remove IS between "participant" and "succesfully"

- line 169: sensor nodes -> sensor node

- line 194: describeD

- line 208: updatING

- line 222: compareD to

- line 226 and 231: where -> while

- line 379: that IS known

- line 380: message IS coming

- line 382: from AN existing

- line 389: of A user

- line 393: belongING to whom or related. Remove also "is" before  belogING

- line 410: that IS only

- line 411: to calculatE

- line 424: by sendING

- line 427: eqauls M9. Remove "with". Also in line 460

- line 439: succesfully be authenticated: remove BE

- line 464: log INTO the system. LET US assume...

- line 469-470: PWi ARE..... an adversary FOR sure cannot..

- line 489: where THERE are lacks... DeatilED explanations.....

- line 490: attacks are already: remove "are"

- line 500: compareD with. Also in line 501

- line 503: The porposed scheme -> This paper

- line 506: updateS

Author Response

The paper proposes a lightweight authentication and key agreement scheme for wireless sensor networks. The authors improve a scheme already proposed by Ali et al. which they claim to be vulnerable to various attacks. Moreover, the authors demonstrate that their method is more lightweight than the one of Ali et al.

The paper seems sound but some doubts remain:

1) first of all: what is the connection with agriculture? why is this method proposed only for agriculture monitoring? It is a quite general scheme for wireless sensor neworks...please explain better or remove the references to the agriculture. Which are the security issues in the agriculture scenario?

Answer: We thank the reviewer for this constructive suggestion.

·       Agriculture is essential for life existence. Such as described in Introduction part, a lot of challenges in agriculture may influence crop productivity and these kinds of risks may affect the world if there is not enough supply from agriculture. With massive development in information technology, it is hoped that agriculture can utilize affordable technology such WSNs and by then may receive such significant assistance in order to face many challenges in agriculture.

·       However, without secure and efficient scheme, the implementation of WSNs in agriculture may bring harm to farmers. Any kind of security issues such as interception-modification-insertion-deletion of important parameters in agriculture can influence the growth, quality and productivity on crops. Therefore, when applying WSNs technology in agriculture, these security issues are very important to be mentioned and noticed.

·       When we see how importance the agriculture for everyone, we know that we need to give a study that focus on it. Other general schemes for WSNs may not necessarily accordance with agriculture monitoring necessities. For example, even for the organization structures such as entities (user, sensor nodes, gateway nodes, cluster head, etc.) can be different with the general schemes because of its different designation purposes. Ali et al. are the first authors whose can see this importance and then designed a special scheme for agriculture monitoring using WSNs. Unfortunately, their scheme suffered from major security weaknesses. This study mentioned their weaknesses and proposed an improved scheme which is more secure and a lot more efficient.

The related descriptions are shown on Pages 1-2, Sec. 1, Paragraphs 1 and 2.

2) literature review is too short, and you cannot group references like this [9-22]: more than 10 references grouped together with no description of their content. Extend the comparison to other key agreement and authentication schemes in peer-to-peer, sensor and distributed networks. I would suggest these references for example:

- https://ieeexplore.ieee.org/abstract/document/5306839

- https://ieeexplore.ieee.org/abstract/document/6364948

- https://ieeexplore.ieee.org/document/7057853

Answer: Thank you for your valuable suggestion. Following the suggestion, we have extended the literature review and give more descriptions about it. The descriptions are shown on Section 1.1.

3) references 24-26 are not described

Answer: Thank you for your valuable suggestion. We have used references [24-26] in Section 1.2 to show that dynamic pseudonym identity schemes are quite popular and widely used in many security researches area.

4) lines 95-100: you do not explain why anonymity is important in an agriculture scenario.

Answer: Thank you for your valuable suggestion. We have revised the descriptions in Section 1.2 about why anonymity is important in agriculture scenario according to the reviewer’s comments.

5) title of section 2 is not appropiate....please select another one.. moreover your paper is only based on an imporvement on Ali et al.' s scheme...it seems a limitation to an incremental study.

Answer: Thank you for your valuable suggestion. We have modified our title for Section 2. We also restructured our manuscript. Additionally, this paper indeed is solely focus on Ali et al.’s scheme’s limitations and its enhancement. Developing a secure and efficient scheme is very important in agriculture monitoring using WSNs. Since Ali et al.’s scheme is such a novel study and we see how important this study is, then this study focus solely to discuss about Ali et al.’s scheme and how to significantly enhance the scheme. Now, we present an improved scheme which is much more efficient and secure if compare with Ali et al.’s scheme.

6) section 2.1: you do not say what is X

Answer: Thank you for your valuable suggestion. We have added information about X in Section 2.2.1 (previously, Section 2.2.1 is Section 2.1). We also added other information to make it clearer for reader when read about system setup phase.

7) section 2.2, step 1: whose are the secret and public keys in the first line of page 4? the user's? you speak about a trustworthy channel: what is that? who guarantees that it is trustworthy?

Answer: Thank you for your valuable suggestion. (1) The step 1 is on user’s side. So, XF is the secret key of user and PF is the public key of user. (2) A trustworthy channel is a confidential and authentic channel, in which transferring data is resistant to overhearing and tampering. A trustworthy channel can be constructed by using physical delivery or using Diffie–Hellman key exchange. We have modified the descriptions and included the explanations on Page 11, Sec. 3.1.

8) what is the advantage of separating BS and GWN? you need more messages and a further entity.

Answer: Thank you for your valuable suggestion. BS acts as system administrator and become the central entity to authenticate other entities. It is because each entity has a shared key with BS. Without BS, Ui, GWNj and SNj will never have chance to truly trust each other.

9) lines 134-136: what about the Sd , you name it and then it is not considered anymore

Answer: Thank you for your thorough review. For section 2.6 (now become section 2.2.6: Dynamic node addition phase), we followed Ali et al.’s scheme annotation where Sd is mentioned and declared in the scheme. But, you are right about Sd where it is only declared once and then is never considered anymore. Therefore, based on your question, we choose to delete the Sd notation.

10) you do not make a comparison between your proposal and Ali et al.'s in terms of number of exchanged messages, are they exactly the same?

Answer: Thank you for your valuable suggestion. Yes, both the proposal and Ali et al.’s scheme have the same number of exchanged messages in login, authentication and key agreement scheme; which is 5 exchanged messages.

11) with which hardware or testbed have you measured the timing in lines 497-498?

Answer: Thank you for your valuable suggestion. The timing line in 497-498 (now become line 543) is following calculation from Ali et al.’s scheme. We do not use any hardware or testbed to measure the timing. Since the proposed scheme is only uses hash function, so, in order to make it easier and clearer, we use computation cost for hash function based on Ali et al.’s scheme.

12) Table 3: which phases are concerned in this time computation? All phases? compirsing also adding of new nodes and change of passwords?

Answer: Thank you for your valuable suggestion. Only login, authentication and key agreement phases are concerned in this computation time. We added information in section 5.2 that the comparisons are only for login, authentication and key agreement phase.

13) have you thought about the security of using only hash operations against a replay attack?

Answer: Thank you for your valuable suggestion. Instead of using only hash operations to against replay attack, we are more confidence to use combination of timestamp and hash operation.

14) Some English typos remain:

- line 24: have -> has

- line 32: plays AN important role...

- line 40: When -> While

- line 41: varying -> vary

- line 42: remains -> remain

- line 48: the people: remove THE

- line 53: integratING

- line 58: improved -> improve

- line 92: which WERE used

- line 64: the data -> data

- line 66: as A basis

- line 70-71: since THEY PLAY AN important role ... systemS and ARE very important (data is plural)

- line 72: data ARE exchanged

- line 144: belongING

- line 145: which user -> a certain user

- line 148: easily -> easy to

- line 151: happened -> happens..... remove IS between "participant" and "succesfully"

- line 169: sensor nodes -> sensor node

- line 194: describeD

- line 208: updatING

- line 222: compareD to

- line 226 and 231: where -> while

- line 379: that IS known

- line 380: message IS coming

- line 382: from AN existing

- line 389: of A user

- line 393: belongING to whom or related. Remove also "is" before  belogING

- line 410: that IS only

- line 411: to calculatE

- line 424: by sendING

- line 427: eqauls M9. Remove "with". Also in line 460

- line 439: succesfully be authenticated: remove BE

- line 464: log INTO the system. LET US assume...

- line 469-470: PWi ARE..... an adversary FOR sure cannot..

- line 489: where THERE are lacks... DeatilED explanations.....

- line 490: attacks are already: remove "are"

- line 500: compareD with. Also in line 501

- line 503: The porposed scheme -> This paper

- line 506: updateS

Answer: Thank you for your valuable suggestion. We have corrected the typos according to the reviewer’s comments.

Round  2

Reviewer 1 Report

The authors have justified most of comments. However, how does this scheme is suitable for agriculture application? I did not see the convincing motivation. 

Through the authors have provided some justification for above, i.e., "Ali, et al. are the first authors whose can see these importance and then designed a special scheme for agriculture monitoring using WSNs. Unfortunately, their scheme suffered from major security weaknesses."

With such justification, I would rather suggest to change the paper title, e.g., Enhanced Lightweight Dynamic Pseudonym Identity Based Authentication and Key Agreement Scheme Using Wireless Sensor Networks for Agriculture Monitoring 

Author Response

Comments and Suggestions for Authors

Reviewer 1:

The authors have justified most of comments. However, how does this scheme is suitable for agriculture application? I did not see the convincing motivation.

 Through the authors have provided some justification for above, i.e., "Ali, et al. are the first authors whose can see these importance and then designed a special scheme for agriculture monitoring using WSNs. Unfortunately, their scheme suffered from major security weaknesses."

 With such justification, I would rather suggest to change the paper title, e.g., Enhanced Lightweight Dynamic Pseudonym Identity Based Authentication and Key Agreement Scheme Using Wireless Sensor Networks for Agriculture Monitoring

Ans.: Thank you for your valuable suggestion. We have revised the descriptions about the importance of wireless sensor network technology for agriculture monitoring and appended the difference between using WSN or P2P networks in agriculture, in healthcare and military purposes, and modified the paper title as “An Enhanced Lightweight Dynamic Pseudonym Identity Based Authentication and Key Agreement Scheme Using Wireless Sensor Networks for Agriculture Monitoring”.

Reviewer 2 Report

The authors tried to respond to my observations but not yet in a fully satisfactory way.

Fig. 1 can be referred to any deployment of sensors and not particularly to an agricultural field. Moreover, in an agricultural field I would suppose sensors are put in lines and not randomly spread.

The answer of the authors to my questions are not sufficient: pleas explain why it is different the deployment of a wireless sensor network in an agricultural field with respect to a forest for example or to an urban environment. Moreover, which particular aspects entail the deployment in agriculture for the security in wireless sensor networks? Please answer these questions otherwise your connection with agriculture is with no meaning.

The reference part is still not sufficient First of all the authors did not consider any of the suggested references herein reported:

- https://ieeexplore.ieee.org/abstract/document/5306839
- https://ieeexplore.ieee.org/abstract/document/6364948
- https://ieeexplore.ieee.org/document/7057853

Then the authors claim that  "Fewer  researchers  discuss  about  authentication and key agreement scheme using WSNs for specific purposes, for example, WSNs for  healthcare through body sensor networks [16, 18, 20], WSNs for military [21] or multimedia [22] or  agriculture monitoring [23]."

But what is the difference between using WSN or P2P networks in healthcare, in agriculture, for military purposes and using WSN or P2P networks in a general, not specific environment? What are the particular and specific issues addressed by these papers in these specific contexts? And consequently, what are the specific issues to be addressed in deploying WSN or P2P networks in an agriculture context?

Lines 110-117 added in this revision simply describe the problem of anonymity in general..but what is the specific problem of anonymity in the agriculture context? There is no answer to my previous question...

The authors say " Developing a secure and efficient scheme is very important in agriculture monitoring using WSNs" but as already said no clues are given about this importance in agriculture compared to other fields..why is more important in agriculture than in a forest or in an urban environment? i.e., what are the specific charachteristics of the agriculture context that affect the security in WSN or P2P networks? At this point, I would question also the novelty and importance of the work of Ali et al. from which you focused so much....

About the trustworthy channel, the authors replied saying that " A trustworthy channel
is a confidential and authentic channel, in which transferring data is resistant to
overhearing and tampering.  A  trustworthy  channel  can  be  constructed  by  using
physical  delivery  or  using  Diffie–Hellman  key  exchange." But DH is vulnerable to a Man-in-the-middle attack and also physicla delivery is not explaianed well in the context of agriculture..how is it performed?

My question "what is the advantage of separating BS and GWN? you need more messages and a further entity" has not been answered properly. You rely on 4 entities as Ali et al. but you do not answer my question: why separating BS and GWN? you need more message exchange in this way

This answer "Thank you for your valuable suggestion. The timing line in 497-498 (now
become line 543) is following calculation from Ali et al.’s scheme. We do not use any
hardware or testbed to measure the timing. Since the proposed scheme is only uses
hash function, so, in order to make it easier and clearer, we use computation cost for
hash function based on Ali et al.’s scheme. " is not good. You cannot replicate experimental data from Ali et al..they are not constant values resulting from a mathemathical computation...timing is computed on a certain hardware and software...

Your answer "Thank  you  for  your  valuable  suggestion.  Instead  of  using  only  hash
operations  to  against  replay  attack,  we  are  more  confidence  to  use  combination  of
timestamp and hash operation." is not correct : "against" cannot be used as a verb alone... Moreover, I would support your statment with some experimental evaluations or simulations.

Author Response

Comments and Suggestions for Authors

Reviewer 2:

The authors tried to respond to my observations but not yet in a fully satisfactory way.

Q1: Fig. 1 can be referred to any deployment of sensors and not particularly to an agricultural field. Moreover, in an agricultural field I would suppose sensors are put in lines and not randomly spread.

Ans.: Indeed. Figure 1 can be referred to any deployment of sensors and not particularly to an agriculture field. This figure is just an illustration for this scheme to make it easier for audiences to understand this WSNs architecture in agriculture.  Also, it does not matter whether sensors are put in lines or randomly spread. This scheme could be utilized to accommodate both of them.

Q2: The answer of the authors to my questions are not sufficient: pleas explain why it is different the deployment of a wireless sensor network in an agricultural field with respect to a forest for example or to an urban environment. Moreover, which particular aspects entail the deployment in agriculture for the security in wireless sensor networks? Please answer these questions otherwise your connection with agriculture is with no meaning.

Ans.: Thank you for your valuable suggestion. We have revised the descriptions and appended the difference between using WSN or P2P networks in healthcare, in agriculture, for military purposes and using WSN or P2P networks and the importance of wireless sensor network technology for agriculture monitoring. Applications used for WSNs in agriculture may use different kinds of sensor devices with ones in healthcare, forest, urban and military since their monitoring range, functionality, power supply status are different. Additionally, wireless sensor network technology for agriculture monitoring can monitor farm temperature, humidity, light, carbon dioxide, soil moisture, acidity, pests, etc., and can be used for plant epidemic monitoring and early warning systems. The descriptions are shown on Page 2, Sec. 1, Lines 54-59.

Q3: The reference part is still not sufficient First of all the authors did not consider any of the suggested references herein reported:

- https://ieeexplore.ieee.org/abstract/document/5306839

- https://ieeexplore.ieee.org/abstract/document/6364948

- https://ieeexplore.ieee.org/document/7057853

Ans.:

We have modified the descriptions and included the suggested references according the reviewer’s comments. The descriptions are shown on Page 3, Sec. 1.1, Lines 94-112.

Q4: Then the authors claim that "Fewer researchers  discuss  about  authentication and key agreement scheme using WSNs for specific purposes, for example, WSNs for healthcare through body sensor networks [16, 18, 20], WSNs for military [21] or multimedia [22] or  agriculture monitoring [23]."

But what is the difference between using WSN or P2P networks in healthcare, in agriculture, for military purposes and using WSN or P2P networks in a general, not specific environment? What are the particular and specific issues addressed by these papers in these specific contexts? And consequently, what are the specific issues to be addressed in deploying WSN or P2P networks in an agriculture context?

Ans.: Thank you for your valuable suggestion. We have revised the descriptions and appended the difference between using WSN or P2P networks in agriculture, in healthcare and military purposes. Applications used for WSNs in agriculture may use different kinds of sensor devices with ones in healthcare, forest, urban and military since their monitoring range, functionality, power supply status are different. Additionally, wireless sensor network technology for agriculture monitoring can monitor farm temperature, humidity, light, carbon dioxide, soil moisture, acidity, pests, etc., and can be used for plant epidemic monitoring and early warning systems.

The descriptions are shown on Page 2, Sec. 1, Lines 54-59.

Q5: Lines 110-117 added in this revision simply describe the problem of anonymity in general..but what is the specific problem of anonymity in the agriculture context? There is no answer to my previous question...

Ans. Thank you for your valuable suggestion. We have revised the descriptions to show the importance of anonymity in agriculture area.

Anonymity is important in agriculture area because it provides privacy and confidentiality by protecting the real identity of participant. In agriculture environment, we can assume that sensor nodes are put openly in the field. If a system does not provide anonymity, an attacker who targeting a particular participant can easily distinguish a transaction belongs to whom. Then he/she is able to perform an attack to his/her particular target. For example, in a farm, the security personnel are responsible for the inspection and monitoring in order to guarantee the security of the farm. If their real identities and daily schedules are exposed and they are tracked, then an adversary can take the opportunity to destroy the farm facilities. It will endanger the safety of the farm.

The descriptions are shown on Pages 3-4, Sec. 1.2, Lines 125-133.

Q6: The authors say " Developing a secure and efficient scheme is very important in agriculture monitoring using WSNs" but as already said no clues are given about this importance in agriculture compared to other fields.. why is more important in agriculture than in a forest or in an urban environment? i.e., what are the specific charachteristics of the agriculture context that affect the security in WSN or P2P networks? At this point, I would question also the novelty and importance of the work of Ali et al. from which you focused so much....

Ans.: Thank you for your valuable suggestion. We have revised the descriptions and appended the importance of wireless sensor network technology for agriculture monitoring. Applications used for WSNs in agriculture may use different kinds of sensor devices with ones in healthcare, forest, urban and military since their monitoring range, functionality, power supply status are different. Additionally, wireless sensor network technology for agriculture monitoring can monitor farm temperature, humidity, light, carbon dioxide, soil moisture, acidity, pests, etc., and can be used for plant epidemic monitoring and early warning systems. The descriptions are shown on Page 2, Sec. 1, Lines 54-59.

Q7: About the trustworthy channel, the authors replied saying that " A trustworthy channel is a confidential and authentic channel, in which transferring data is resistant to overhearing and tampering.  A trustworthy channel can be constructed by using physical delivery or using Diffie–Hellman key exchange." But DH is vulnerable to a Man-in-the-middle attack and also physicla delivery is not explaianed well in the context of agriculture.. how is it performed?

Ans.: Thank you for your valuable suggestion. We have modified the user/agriculture professional registration phase such that the proposed scheme does not require a trustworthy channel. Each user Ui has a SC which contains a pre-configured identity IDipre and a random number r0. The pre-configured data is also stored in BS’s storage. The SC is transferred by using physical delivery. Then Ui  and BS can securely deliver registration messages without assumption of a trustworthy channel. The detailed descriptions are shown on Page 12, Sec. 3.1 and Figure 5.

Q8: My question "what is the advantage of separating BS and GWN? you need more messages and a further entity" has not been answered properly. You rely on 4 entities as Ali et al. but you do not answer my question: why separating BS and GWN? you need more message exchange in this way

Ans.: BS and GWN have different functions. BS acts as a trusted server. It regulates key exchanges and accommodates trustworthiness between Ui, GWNs and SNs. Practically it connects Ui with particular GWNs. On the other side, GWNs acts as a local server for a group of sensor nodes. It works locally for particular sensor nodes. Logically, a farmer requires its own GWNs. Otherwise he/she may be able to access other groups of sensor nodes that belongs to others and vice versa. That’s why the separation between BS and GWN is necessary. Even it may look that this scheme needs more message exchanges because of separation between BS and GWN, we still believe that it’s the best to separate BS and GWN because of their functions are different in this scheme.

Q9: This answer "Thank you for your valuable suggestion. The timing line in 497-498 (now become line 543) is following calculation from Ali et al.’s scheme. We do not use any hardware or testbed to measure the timing. Since the proposed scheme is only uses hash function, so, in order to make it easier and clearer, we use computation cost for hash function based on Ali et al.’s scheme. " is not good. You cannot replicate experimental data from Ali et al.. they are not constant values resulting from a mathemathical computation...timing is computed on a certain hardware and software...

Ans.: We have revised the descriptions about performance comparisons and included the simulation according reviewer’s comments. The descriptions are shown on Page 23, Sec. 5.2, Table 3 and Table 4.

Q10: Your answer "Thank you for your valuable suggestion.  Instead  of  using  only  hash operations  to  against  replay  attack,  we  are  more  confidence  to  use  combination  of timestamp and hash operation." is not correct : "against" cannot be used as a verb alone... Moreover, I would support your statment with some experimental evaluations or simulations.

Ans.: Thank you for your valuable suggestion. Indeed, an authentication and key agreement scheme may be developed by using only hash operations, and is secure against replay attacks. However, the scheme will increase communication overhead because of challenge/ response technique. Besides, we have revised the descriptions about performance comparisons and included the simulation according reviewer’s comments. The descriptions are shown on Page 23, Sec. 5.2, Table 3 and Table 4.

Round  3

Reviewer 2 Report

Ans.:  Indeed.  Figure  1  can  be  referred  to  any  deployment  of  sensors  and  not
particularly to an agriculture field. This figure is just an illustration for this scheme to
make  it  easier  for  audiences  to  understand  this  WSNs  architecture  in  agriculture.   
Also,  it  does  not  matter  whether  sensors  are  put  in  lines  or  randomly  spread.  This
scheme could be utilized to accommodate both of them.

If this is a general deployment you should not state in the caption " Agriculture monitoring system model of the proposed scheme". Moreover, I suggest to put sensor in rows if you consider agriculture deployment, because usually plants are put in rows in an agriculture field.

Ans.: Thank you for your valuable suggestion. We have revised the descriptions and
appended  the  difference  between  using  WSN  or  P2P  networks  in  healthcare,  in
agriculture, for military purposes and using WSN or P2P networks and the importance
of wireless sensor network technology for agriculture monitoring. Applications used
for  WSNs  in  agriculture  may  use  different  kinds  of  sensor  devices  with  ones  in
healthcare,  forest,  urban  and  military  since  their  monitoring  range,  functionality,
power supply status  are  different. Additionally,  wireless sensor network technology
for  agriculture  monitoring  can  monitor  farm  temperature,  humidity,  light,  carbon
dioxide,  soil  moisture,  acidity,  pests,  etc.,  and  can  be  used  for  plant  epidemic
monitoring and early warning systems. The descriptions are shown on Page 2, Sec. 1,
Lines 54-59.

You did not provide a difference between agriculture and forest or urban environments. Are sensors put in a different way in agriculture compared to cities or forests? Also in forests or cities sensors can sense temperature,  humidity,  light,  carbon dioxide,  soil  moisture,  acidity,  pests, etc.

Ans. Thank you for your valuable suggestion. We have revised the descriptions to
show the importance of anonymity in agriculture area.   
Anonymity  is  important  in  agriculture  area  because  it  provides  privacy  and
confidentiality  by  protecting  the  real  identity  of  participant.  In  agriculture
environment, we can assume that sensor nodes are put openly in the field. If a system
does  not  provide  anonymity,  an  attacker  who  targeting  a  particular  participant  can
easily distinguish a transaction belongs to whom. Then he/she is able to perform an
attack to his/her particular target. For example, in a farm, the security personnel are responsible for the inspection and monitoring in order to guarantee the security of the
farm. If their real identities and daily schedules are exposed and they are tracked, then
an adversary can take the opportunity to destroy the farm facilities. It will endanger
the safety of the farm.
The descriptions are shown on Pages 3-4, Sec. 1.2, Lines 125-133.

Anonymity is different from privacy and confidentiality: you should clear your mind about these three different concepts and decide what you want to say and WHY this is important in agriculture. Anonymity regards the protection of the identities of the parties involved in a communication. Why knowing that Alice is receiving data from a humidity sensor is important to be protected? or why knowing that Alice is communicating with the sensors can harm the farm facilities? in which way?

Ans.: We have revised the descriptions about performance comparisons and included
the simulation according reviewer’s comments. The descriptions are shown on Page
23, Sec. 5.2, Table 3 and Table 4.

You have added the hardware specifications in table 4 but you did not change the data of table 3..how is it possible? you said you replicated the data of Ali et al. but if you have performed tests on your own hardware the data of table 3 should change, shouldn't they?

Minor remarks:

line 110:  which  shared  key -> WHOSE shared key

and some minor English typos

Author Response

Comments and Suggestions for Authors

Q1.:

If this is a general deployment you should not state in the caption " Agriculture monitoring system model of the proposed scheme". Moreover, I suggest to put sensor in rows if you consider agriculture deployment, because usually plants are put in rows in an agriculture field.

Ans.: Thank you for your valuable suggestion. We have revised Figure 1 in Page 2 such that sensors are put in rows for agriculture deployment.

Q2.:

You did not provide a difference between agriculture and forest or urban environments. Are sensors put in a different way in agriculture compared to cities or forests? Also in forests or cities sensors can sense temperature, humidity, light, carbon dioxide, soil moisture, acidity, pests, etc.

Ans.: Thank you for your valuable suggestion. We have revised the descriptions and included differences about the deployment and function of sensors between forests, military, urban areas and agriculture. For the deployment, the sensors in forests and military must consider the terrain such as rivers, valleys, etc. and are put in irregular. The distribution of urban sensors must consider factors such as roads and buildings. The sensors of the human body must consider the body shape and portability. The sensors in agriculture are often arranged in row due to the arrangement of crops. In terms of function, the sensors in forests and farms sense temperature, humidity, light, carbon dioxide, soil moisture, acidity, pests, etc. The urban sensors sense dust, air pollution, temperature, humidity, etc. In addition to sensing sound and images, sensors in military must sometimes be able to sense toxic or chemical substances.

The descriptions are shown on Page 2, Sec. 1, Lines 59-65.

Q3.

Anonymity is different from privacy and confidentiality: you should clear your mind about these three different concepts and decide what you want to say and WHY this is important in agriculture. Anonymity regards the protection of the identities of the parties involved in a communication. Why knowing that Alice is receiving data from a humidity sensor is important to be protected? or why knowing that Alice is communicating with the sensors can harm the farm facilities? in which way?

Ans.: Thank you for your valuable suggestion. We have revised the descriptions to explain the importance of anonymity in agriculture area.

Anonymity is important in agriculture area because it provides legitimate users with protection of their real identities. In agriculture environment, we can assume that sensor nodes are put openly in the field. If a system does not provide anonymity, an attacker who targeting a particular participant can easily distinguish a transaction belongs to whom. Then he/she is able to perform attacks to his/her particular target. For example, Alice is an epidemic specialist and works on a farm. An adversary who tries to harm the farm facilities obtains Alice's identity and knows that she is responsible for assisting in monitoring the farm's temperature, humidity, and pests. The adversary may perform social engineering or dictionary attacks to obtain Alice's password or login information, and then can login the system for agriculture monitoring to tamper with information and damage facilities.

The descriptions are shown on Page 4, Sec. 1.2, Lines 131-140.

Q4.:

You have added the hardware specifications in table 4 but you did not change the data of table 3..how is it possible? you said you replicated the data of Ali et al. but if you have performed tests on your own hardware the data of table 3 should change, shouldn't they?

Ans.: For the computational cost, Ali et al.’s scheme requires 19 hash function and 9 symmetric en/decryption operations, and the proposed scheme requires 37 hash function operations. The computational cost required in each scheme is constant. However, the response time is depend on the hardware specifications, and will be changed. For example, the total response time of Ali et al.’s scheme is 1.1803ms. and in Ali et al.’s hardware specifications. But the total response time of Ali et al.’s scheme is 0.03512 sec.(=35.12ms.), and the total response time of the proposed scheme is 0.00740 sec. (= 7.40ms.) in our hardware specifications.

The descriptions are shown on Page 23, Sec. 5.2, Table 3 and Table 4.

Minor remarks:

Q5.:

line 110:  which  shared  key -> WHOSE shared key

and some minor English typos

Ans.: Thank you for your valuable suggestion. We have checked the manuscript and revised some minor English typos. The descriptions are shown on Page 2, Line 55 and Line 66; Page 3, Line 117; and on Page 10, Line 243.
